# Can TROPOMI-NO$_2$ satellite data be used to track the drop and resurgence of NO$_x$ emissions within Germany between 2019 - 2021 using the multi-source plume method (MSPM)?

Enrico Dammers[1], Janot Tokaya[1], Christian Mielke[2], Kevin Hausmann[2], Debora Griffin[3], Chris McLinden[3], Henk Eskes[4], and Renske Timmermans[1]

[1]Netherlands Organisation for Applied Scientific Research (TNO), Utrecht, Princetonlaan 6, 3584 CB, The Netherlands
[2]Umweltbundesamt (UBA), Dessau-Roßlau, Wörlitzer Pl. 1, 06844, Germany
[3]Air Quality Research Division, Environment and Climate Change Canada, 4905 Dufferin St, M3H 5T4, Toronto, Canada
[4]KNMI, De Bilt, Utrechtseweg 297, 3731 GA, The Netherlands

**Correspondence:** Enrico Dammers (enrico.dammers@tno.nl)

**Abstract.** NO$_x$ is an important primary air pollutant of major environmental concern which is predominantly produced by anthropogenic combustion activities. NO$_x$ needs to be accounted for in national emission inventories, according to international treaties. Constructing accurate inventories requires substantial time and effort, resulting in reporting delays of one to five years. In addition to this, difficulties can arise from temporal and country specific legislative and protocol differences. To address these issues, satellite-based atmospheric composition measurements offer a unique opportunity for independent and large-scale estimation of emissions in a consistent, transparent, and comprehensible manner. Here we test the multi-source plume method (MSPM) to assess the NO$_x$ emissions over Germany in the Corona period from 2019-2021. For the years where reporting is available, the differences between satellite estimates and inventory totals were within 75-100 kt (NO$_2$) NO$_x$ (<10% of inventory values). The large reduction of NO$_x$ emissions ($\sim$15%) simultaneous with the COVID-19 lock-downs was observed in both the inventory and satellite derived emissions. The recent projections for the inventory emissions of 2021 pointed to a recovery of the 2021 emissions towards pre-COVID-19 levels. In the satellite derived emissions however, such an increase was not observed. While emissions from the larger power plants did rebound to pre-COVID-19 levels, other sectors such as road transport did not, likely due to a reduction in the number of heavier transport trucks. This again illustrates the value of having a consistent satellite based methodology for faster emission estimates to guide and check the conventional emission inventory reporting. The method described in this manuscript also meets the demand for independent verification of the official emission inventories, which will enable inventory compilers to detect potentially problematic reporting issues, bolstering transparency and comparability: two key values for emission reporting.

## 1 Introduction

Nitrogen monoxide (NO) and dioxide (NO$_2$) play an important role in the atmospheric chemistry as they influence the abundance of tropospheric ozone (Seinfeld and Pandis, 2006), and lead to aerosol formation. These primary air pollutants are collectively called nitrogen oxides (NO$_x$ $\equiv$ NO + NO$_2$). Since NO$_2$ is for the most part formed primarily through rapid ox-

idation of NO, their concentrations are strongly related. $NO_2$ is a major source of air pollution and exposure can result in significant health problems that cause an association between long-term exposure and reduced life expectancy (Atkinson et al., 2018; Belch et al., 2021). Hence, objective concentration limits are set by the European Union on the hourly (200 $\mu$g m$^{-3}$)

and yearly (40 $\mu$g m$^{-3}$) $NO_2$ exposure levels, with recent World Health Organization (WHO, 2021) limits reducing the annual mean limit to 10 $\mu$g m$^{-3}$. As well as adverse health effects, $NO_2$ also places a strain on the environment through soil and water acidification as well as eutrophication (Galloway et al., 2003).

Many anthropogenic activities contribute to the atmospheric $NO_2$ concentration since $NO_2$ is formed in combustion processes where air (being about 80% nitrogen) is the oxidant. Natural sources of $NO_x$ include lightning and soil emissions. The

main sources of $NO_x$ emissions are the internal combustion engines that burn fossil fuels in motor vehicles and industry. The overall atmospheric evolution and budget of $NO_x$ in the atmosphere has been determined with ever-increasing accuracy over the last decades. National environmental agencies are required to monitor the level of $NO_x$ and the contribution of human activity on it according to international agreements, such as the Convention on Long-Range Transboundary Air Pollution (CLRTAP, https://unece.org/environment-policy/air) by the United Nations Economic Commission for Europe (UNECE). Efforts under-

taken to limit $NO_x$ emissions have resulted in strong reductions of ambient $NO_2$ concentration in many parts of the world (Jamali et al., 2020).

Inventories of $NO_x$ emissions are commonly compiled using a bottom-up approach based on proxies as well as direct emission measurements, for example in stacks. Retrieving data at detailed levels as well as creation of representative emission factors that translate an activity into emissions is, however, a very labour intensive task. For example, emissions from road

transport depend on several factors such as: fleet composition, type of fuel, engine maintenance and design, outside temperature, usage profile and road conditions. New technology standards, reported numbers, and real life measures (or lack thereof compared to emission estimates) are slow to be incorporated in the emission inventories, as they need to fulfill the good practice guidelines of the respective protocol commonly agreed upon by the EU member states. Therefore, inventories cannot reflect the latest actual emission trends in "near-real-time". This is problematic especially when large deviations from business as usual

scenarios occur, which are then only reflected in the inventories with a great time lag. For example, air quality forecasts depend on accurate emission inventories to represent these changes. A recent example is the large changes in emissions following the COVID-19 lock-downs and the post lockdown recovery phase of the emissions, which both are poorly represented in current air quality applications (Goldberg et al., 2020; Griffin et al., 2020; Barré et al., 2021).

A potential solution to speed up the creation of up to date emissions from inventories, in a harmonized way, is the usage

of satellite observations of air pollutants (Beirle et al., 2011; Fioletov et al., 2011; Mijling et al., 2009; Miyazaki et al., 2012; McLinden et al., 2016; Goldberg et al., 2019; Griffin et al., 2020; Dammers et al., 2019; Ding et al., 2020), which can be used to verify the reported emissions, constrain emission sources, and analyse trends. Furthermore, methods that allow for independent verification, can potentially be used to trace and reveal significant discrepancies in the current emission inventories and have proven to be accurate. An example would be the Diesel-Gate Scandal (Jonson et al., 2017), which revealed that diesel cars had

been emitting at least four times more $NO_x$ in on-road driving than in type approval tests. Timely verification of the inventories could potentially identify such discrepancies more rapidly.

Over the past decade the data availability of satellite-based atmospheric composition measurements has increased tremendously. Furthermore, due to increased instrument sensitivity and spatial and temporal resolution, these satellite-based measurements are becoming more and more attractive for air quality monitoring and emission studies. Recent scientific developments have shown the viability of various methods in estimating emissions based on satellite observations. In the case of $NO_x$, the earliest methods were mostly developed to estimate the emissions of individual point sources (Beirle et al., 2011) followed by regional estimates at lower spatial resolutions (Mijling et al., 2009; Miyazaki et al., 2012). The more recent TROPOspheric Monitoring Instrument (TROPOMI) with its unprecedented spatial resolution of $3.5 \times 5.5\,km^2$ improved the resolvability of individual and clusters of emission sources (Ding et al., 2020; McLinden et al., 2022).

The TROPOMI-$NO_2$ product offers an inventory independent source to verify $NO_x$ emissions. These observations of spatio-temporal trends offer the possibility for inventory agencies to independently check their findings on, for example, emission reduction of $NO_x$ throughout the country without having to rely on bottom-up inventory data products, such as the Emission Database for Global Atmospheric Research (EDGAR) (Crippa et al., 2019), the Copernicus Atmospheric Monitoring Service (CAMS) database (Granier et al., 2019; Kuenen et al., 2022), or other country specific gridded data products like from the Gridding Emission Tool for ArcGis (GRETA) (Schneider et al., 2016). Fast changing spatio-temporal patterns may only be captured by space-borne data in a timely manner in comparison to the above mentioned gridded data products.

A major driver behind the research work presented here is the provision of a tool, developed for the Umweltbundesamt (UBA, German Environment Agency), to compare satellite-derived emissions with inventory emissions for air pollutants, in order to verify the bottom-up computed emissions with independent data from space-borne measurements. This should help inventory compilers to build trust in their work and identify potentially problematic issues, in case large deviations between inventory data and space-borne data trends are present. Furthermore, the tool should allow for fast checks if a country is compliant with its national air pollutant reduction targets, which have been initiated by the EU (EU, 2022) or if adjustments need to be made (Dore, 2022).

In this study, we apply one of the more recently developed methods (Fioletov et al., 2017) to TROPOMI-$NO_2$ observations to derive the $NO_x$ emissions for Germany for the period of 2019-2021. The plume based fitting method relies on wind data and a parameterization of multiple Gaussian plumes originating from corresponding point sources to estimate the strength of the emissions at these point source location. These estimates are then compared to the emissions in the current inventories for 2019 and 2020, as well as the projected emissions of 2021 to assess their validity and analyse the expected variations due to the COVID-19 lock-downs. The plume based fitting routine is part of an open-access standalone tool[1]. Besides the plume based fitting routine two additional methods were implemented during the development phase, a simple mass-balance approach for which we coined the term "naive method", and the divergence approach as described by Beirle et al. (2019). Furthermore, the simple mass-balanced method was employed into an online web-tool [2], geared towards emission inventory agencies that are interested to compare their national total emissions to an independent, yet easily comprehensible space-borne emission

---

[1]https://github.com/UBA-DE-Emissionsituation/space-emissions

[2]https://space-emissions.net/

estimate. More details on the implementation and comparison of these methods can be found in Dammers et al. (2023). In this
study, we focus on the results of the plume-based fitting method.

## 2 Methodology and Datasets

### 2.1 Data sets

#### 2.1.1 Emission inventory

The reporting of the national air pollutants follows international guidelines that are available via the EMEP centre on Emission
Inventories and Projections[3]. The reported inventory data for Germany, in form of the detailed informative inventory report
(IIR), describing the technical methodology, may be found here:
https://iir.umweltbundesamt.de/2022/.

The data is arranged in time series per gas species considering the different emission sources of $NO_x$ in sectors such as: e.g.
public power, industry and traffic in very detailed, disaggregated form, at the national level. The bottom up creation of these
inventories is driven by statistical data provided by the German statistical authority (DESTASIS), and complex models that use
this data to compile the emissions for a specific gas (or aerosol) for a specific emission source in a specific sector and year.
The uncertainties for each reported emission source depend on the availability of the data used for the emission calculation,
and may vary considerably. As an example, uncertainties in emissions from sectors, which are quite accurately described by
statistical data and models such as emissions from large power plants, show much lower uncertainties than sectors that are
governed by a great complexity such as the natural variability of $NO_x$ from agricultural emissions in soils (e.g., uncertainties
can be more than 300 % for agricultural soils, UBA (2023)).

In this study both the gridded (CLRTAP, 2021) and non-gridded (CLRTAP, 2022) reported emission data sets are retrieved
directly from the respective Convention on Long-range Transboundary Air Pollution (CLRTAP) inventories, which follow
the nomenclature for reporting (NFR) standard. The gridded data set is only available for 2019, whilst the non-gridded data
is available for both 2019 and 2020. The 2021 data is a prognosis based on the trends observed between 2012-2019 for all
emission classes, under the assumption that the patterns in most emission sectors rebound after the 2020 COVID-19 lock-
downs. An overview of relative contributions of individual sectors to the total gridded emissions is shown by sector in Fig.
A1.

All emissions except the MEMO items (MEMO items are additional reported emissions on non-standard emission such
as volcanoes, forest fires etc.) are selected from the CLRTAP inventories. Two natural sources are added to this set, namely
non-agricultural soils and lightning. Globally, the lightning NO constitutes about 3% of the total $NO_x$ emission budgetEEA
(2019). According to the guidebook (EEA, 2019) only 20% of the lightning NO is formed in the lowest 1000 meter of the
atmosphere and the remaining 80% at higher altitude (all inter cloud lightning above 5 km height. A rough estimate for the
lightning emissions can be made on the basis of the number of flashes per $km^2$ and the expected $NO_x$ emissions released per

---

[3]https://www.ceip.at/reporting-instructions

flash. A study by Anderson and Klugmann (2014) gives an average of about 2 flashes per km$^2$ throughout Germany, with fewer flashes in the central and northern parts. Assuming that on average the number of lightning flashes did not increase significantly in combination with a production of about 180 mol NO/flash (Bucsela et al., 2019), and German surface area of about 357,000 km$^2$, gives us a German lightning NO$_x$ emission total of about 5 kt (NO$_2$) NO$_x$ per year. This emission total is very minor and spread out over a large domain, and not expected to be a significant source of error when comparing the satellite derived emission estimates with the emission inventory. From this point forward in the manuscript kt (NO$_2$) NO$_x$ is written as kt NO$_x$.

Due to widespread nitrogen pollution and deposition in Germany it is complicated to make an estimate of purely non-anthropogenic and non-agricultural soil emissions. There are several studies that looked at soil NO$_x$ emissions for the European domain, mostly based on the anthropogenic emissions Yienger and Levy (1995) reported, but fewer that focus on purely natural emissions. Simpson et al. (1999) gave an estimate of 3-90 kt NO$_x$ of forest emissions and 20 kt NO$_x$ for grassland soils. This estimate was more recently updated by Simpson and Darras (2021) and is available as the CAMS-GLOB-SOIL inventory

(Simpson, 2022), with a reported 2018 German emission total of about 160 kt NO$_x$. Within the inventory the emissions are split between fertilizer induced, biome, deposition related, and pulsed soil emissions. There is always a danger of double counting such emissions but the fertilizer induced emissions of 100 kt NO$_x$ match closely to those included within the 2019 GNFR data of approximately 110 kt NO$_x$ (classed under L. AgriOther sources). The remaining 60 kt of NO$_x$ per year is a combination of biome, deposition related, and pulsed soil emissions. The distribution of non-agricultural source emissions are

quite uniformly distributed throughout Germany, peaking somewhat towards the northeastern part of the country. Note that Simpson and Darras (2021) stress that the derived soil emissions still have a large uncertainty range, mostly related to a lack of observations, missing data for some biomes, and the uncertainty of input parameters such as soil temperatures. Annual variations are expected to be large depending on variations in soil temperatures. Simpson and Darras (2021) do not provide an

upper and lower range of the emissions.

Additionally, we use the European Pollutant Release and Transfer Register (E-PRTR) for the emission locations and strengths of the largest industrial emission sources within Germany. The latest data set (v18) can be accessed via: https://www.eea.europa.eu/data-and-maps/data/industrial-reporting-under-the-industrial-6. Only sources with an emission strength above 0.25 kt NO$_x$ per year are selected for later comparison to the satellite derived emissions. Note, that the most recent data available is based on reported

emissions of the year 2017, and thus we only use the data as a rough indication of source strength.

### 2.1.2 TROPOMI-NO$_2$

The TROPOMI instrument, on the S5P satellite platform, was launched on the 13[th] of October 2017. The satellite instrument achieves almost full daily coverage of the globe through a sun-synchronous orbit with a local overpass at around 13:30 (Veefkind et al., 2012). TROPOMI has an unprecedented horizontal resolution of $3.5 \times 5.5 \, \text{km}^2$ for the NO$_2$ product. Details

on the retrieval are described in the Copernicus user manuals (Algorithm Theoretical Basis Document (ATBD)) as well as in earlier publications such as van Geffen et al. (2022). The TROPOMI-NO$_2$ operational product has three data streams. The near-real-time product available within 3 hours (NRTI), the offline (OFFL) version that follows one day later and receives a more stringent quality control (now spanning 2019-2021) and a complete reprocessed version that is provided at more irregular

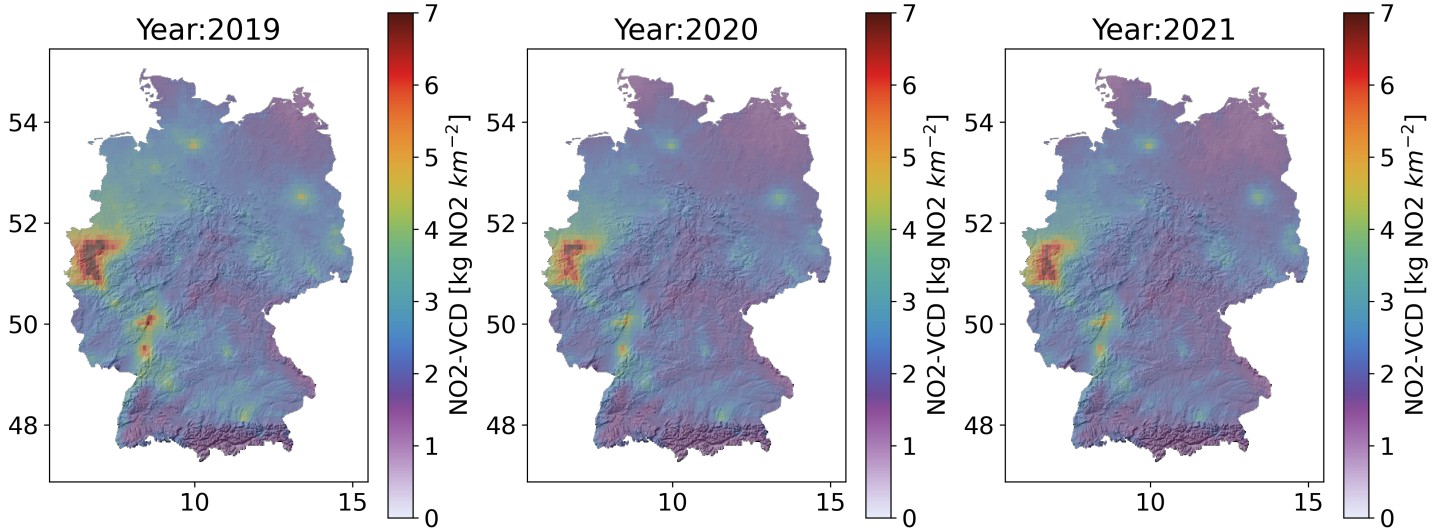

**Figure 1.** TROPOMI-NO$_2$ (PAL product, v2.3.1) year averaged vertical column density concentrations over Germany for the years 2019-2021.

intervals (RPRO, April 2018-November 2018). Over time, several improvements in the retrieval algorithm lead to processor
updates and new product versions. Finally, independently from the operational steams, a reanalysis of the full data set with the most up-to-date retrieval algorithm became available at the end of 2021, named the PAL product, which is currently available until the end of November 2021, connecting seamlessly to OFFL v2.3.1 from November 2021 to July 2022. The TROPOMI-NO$_2$ product went through several upgrades concerning its product versions over the years, with the most recent three upgrades from version v1.3.2 to version v1.4.0, then v2.2.0, then v2.3.1, and v2.4.0 taking place in respectively November 2020, July
2021, November 2021 and July 2022. The most recent upgrade to version 2 involved a more major overhaul greatly improving the overall quality of the retrieval (van Geffen et al., 2022; Zhao et al., 2022).

The TROPOMI-NO$_2$-PAL product includes a reanalysis of the earlier data and provides a consistent version throughout (v2.3.1). This product is recommended to be used for any longer time series analysis and has been used in this study. We combine this product with 2 months of the newest OFFL data (v2.3.1) to complete the data series for 2021. The PAL product
is available through the PAL data portal [4].

The quality of the TROPOMI-NO$_2$ PAL and OFFL products based on the v2.3.1 processor version is discussed by van Geffen et al. (2022). Furthermore, the previous data set versions 1.2.x and 1.3.x were relatively well validated (Verhoelst et al., 2021). The TROPOMI NO$_2$ data correlate well when compared to ground-based MAX-DOAS and PANDORA instruments (Verhoelst et al., 2021), but tend to show an underestimation of the tropospheric column. The median negative bias ranges
from -15% to -35% in most clean to slightly polluted regions and up to -50% over highly polluted regions for versions 1.2.x and 1.3.x. This bias is reduced in the PAL data set (van Geffen et al., 2022) with reported improvements for the tropospheric columns from an average low bias of -32% to -23%. The range of the differences for individual sites are, however, quite wide

---

[4]https://data-portal.s5p-pal.com/products/no2.html

with for example the MAX-DOAS in de Bilt, the Netherlands, showing a range of around -75% up to around +50% (25 and 75 percentiles) with a median of around -20%. The negative bias can be explained by the low spatial resolution of the a-priori profiles as well as the treatment of clouds and aerosols in the retrievalLange et al. (2023). As for the TROPOMI data quality criteria, the requirements recommended in the ATBD were used, which means observations with a cloud fraction below 0.03 were used, based on the `cloud_fraction_crb_nitrogendioxide_window` variable in the data files. Furthermore, observations with a quality value (`qa_value`) below 0.75 were filtered from the data set. It is important to note that the MAX-DOAS and PANDORA instruments are not completely free of bias themselves, however the ground-based instruments typically have much lower uncertainties than the TROPOMI-NO2 product as stated in Verhoelst et al., 2021.

Figure 1 shows yearly averages of the TROPOMI-NO$_2$ (PAL, v2.3.1) data. Here the reduced column denisities that occur simultaneous with the COVID-19 lockdown measures in 2020 is clearly visible: The industrialized Ruhr valley at the western border of Germany shows far reduced levels of NO$_2$, if compared to 2019. The same is also observed in the industrial centers further to the South-South-West, which almost vanishes in 2020 and show only a very slow recovery of emissions in 2021.

### 2.1.3 Wind data

The methodology in this study makes use of the wind rotation approach as explained in detail in Pommier et al. (2013), Fioletov et al. (2015), and Dammers et al. (2019). The required wind data is taken from ECMWF's ERA-5 dataset (Hersbach et al., 2020, 2018), which was downloaded at a 0.25°x0.25° resolution and 1-hour temporal resolution. To match each of the satellite footprints the meteorological fields are interpolated (spatially and temporally) to each of the observations. We assume that the majority of the NO$_x$ mass from local emissions is located in the lower boundary layer (Beirle et al., 2019; McLinden et al., 2022; Griffin et al., 2020) and for the transport of NO$_x$ an average is taken of the wind fields of the first 100 hPa (around the first kilometre) above the surface. These are approximately the levels between ~1000-900 hPa for a typical sea level location, and for a location with a surface pressure of 800 hPa, winds between 800 and 700 hPa are averaged. The surface pressure at the location of the satellite observations are used to determine the 100hPa layer.

### 2.2 Emission estimation tool

The plume based fitting routine presented here was developed together with two other methods to form the core of the satellite-based emission tool as developed for the UBA. The other two methods are a simple mass-balance approach, which was coined the "naive method", and a third method, the divergence approach as described by Beirle et al. (2019). The tooling is available in two forms, the aforementioned open-access standalone offline tool ()https://github.com/UBA-DE-Emissionsituation/space-emissions, and an online web-tool. The focus in this study is on the plume based fitting method. More details on the other methodologies and inter-comparison of these other methods can be found in Dammers et al. (2023). The tool is offered as a web-based application available at: https://space-emissions.net/. The data processing is hosted at the German national Copernicus data service initiative[5], which offers a direct link to the required Sentinel-5P data products especially tailored for governmental agencies of Germany. This web-based application tool is directly targeted at users from the emission inventory community

---

[5]https://code-de.org/de/

and, therefore, uses the TEMIS monthly L3 data product available at https://www.temis.nl/. The design of the tool is based on a modular structure that encourages later additions of other compatible air pollutants to the tool chain such as $SO_2$ or to add more technical and more computationally demanding methods, other than the mass-balance technique employed currently in the online tool, in later development steps. This was necessary as it offers a concise development framework to which more advanced techniques may be added later on, driven by the emission inventory community.

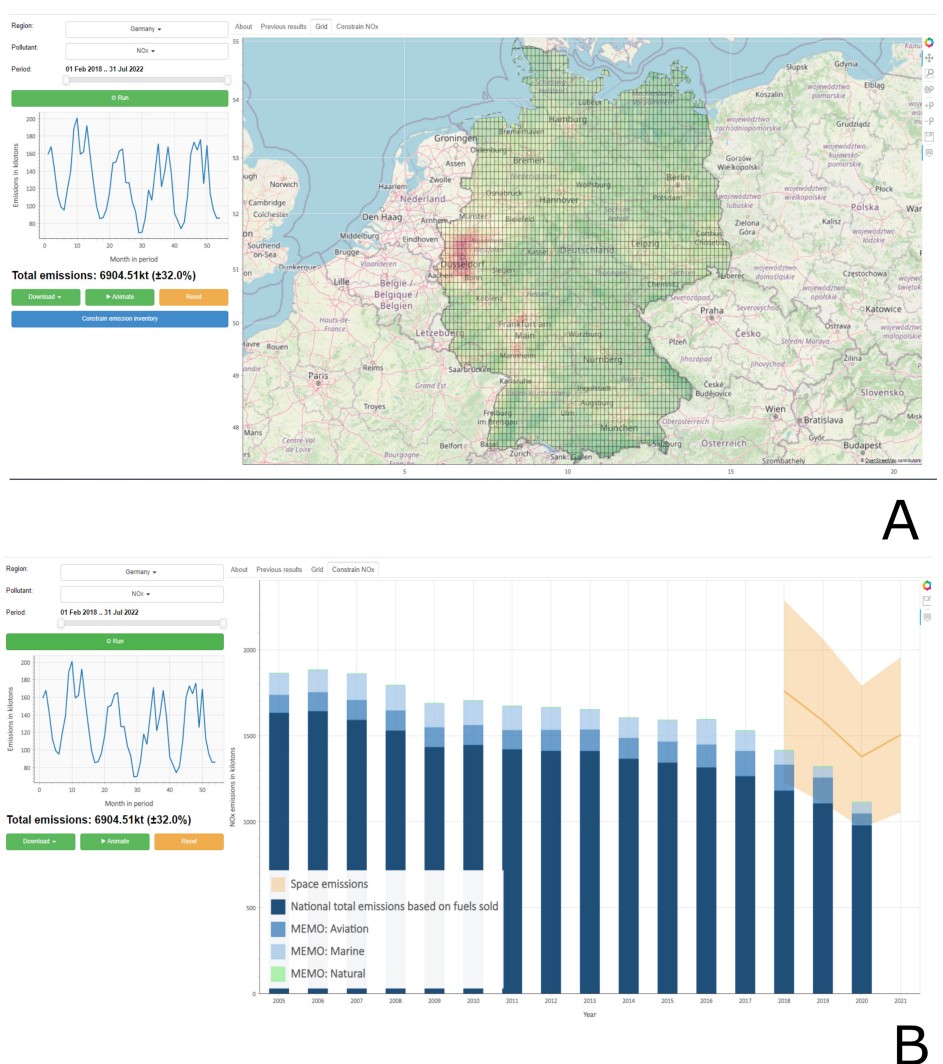

**Figure 2.** Screenshots from the satellite-based emissions tool https://space-emissions.net/ for Germany. Panel A illustrates the interface and the visualisation of the result. Whist panel B illustrates the result in context to the reported $NO_x$ values.

The online tool works as follows, (also shown in Fig. 2): a user is required to select the country of the world they want to target with their analysis, as well as the desired air pollutant (in this case $NO_2$), from the respective pull down menus of the processing options (online) or by providing a shapefile of the region of choice (offline). After that, the time span for the observations needs to be selected, covering the period for which the data is available. The user initiates the computation, which returns the analysis results to the graphical user interface. The user may then download the graphical results as well as the analysis results as

a comma separated value (.csv) file and/or other ancillary data using the post-processing options (netCDF4 files). Advanced users and software developers are also encouraged to visit: https://github.com/UBA-DE-Emissionsituation/space-emissions for the source code of the project.

### 2.2.1   Multi-source plume method (MSPM)

Emissions were derived using the Multi-source plume method (MSPM) (Fioletov et al., 2017; McLinden et al., 2022; Dammers

et al., 2022), which was originally developed by Fioletov et al. (2017) and can be used for an assessment of emissions from both area and point sources. For a more detailed explanation we refer readers to those publications. In short, the method relates observations and emission sources by creating a linear system of plume functions, which effectively establish a system of source and receptor relations in which the total tropospheric column density of each observation is described as a combination of the total column densities of all sources plume functions. This is expressed as:

$$\mathbf{Ax} = \mathbf{B} \qquad\qquad (1)$$

where $\mathbf{A}$ is the linear system of source-receptor relations, $\mathbf{x}$ the emission sources, and $\mathbf{B}$ the satellite observed vertical column density (in our case the TROPOMI-$NO_2$ tropospheric columns). Several additional terms can be incorporated in $\mathbf{x}$ to account for regional product biases and for background concentrations. While the TROPOMI-$NO_2$ product does have local biases, the small number of validation stations hampers an accurate determination and correction for the product bias. To account

for the bias we apply a correction on an overpass to overpass basis, following Beirle et al. (2019), removing the lowest 5% of the observed total column density within the larger domain. The short lifetime of $NO_2$ ensures that further corrections for background concentrations are not needed.

Any plume function can be used to represent the relations in matrix $\mathbf{A}$, here we use the exponentially modified Gaussian (EMG) plume function, which has been successfully applied in previous studies (de Foy et al., 2014; Fioletov et al., 2017;

McLinden et al., 2020; Dammers et al., 2019). Using this method, observations are rotated around a single point, the emission source, so that each is positioned in a similar upwind-downwind frame (Pommier et al., 2013) with respect to the wind direction. This enables us to describe the position of each observation as a point within a downwind plume. For more details on the plume rotation method see, Fig. S4 in Pommier et al. (2013). The EMG plume function describes the vertical column density (VCD) concentrations downwind of a source. The VCD at each position x,y near a source can be described by Eq. 2, where *a* represents

the emission enhancement, *f(x,y)* the crosswind diffusion (Eq. 4), and *g(y,s)* (Eq. 6) a convolution of the downwind advection and diffusion. Within all functions *x* represents the crosswind position, *y* the downwind position, and *s* the wind-speed.

$$V(x,y,s) = a \cdot f(x,y) \cdot g(y,s) + B \tag{2}$$

$$\sigma_1 = \begin{cases} \sqrt{\sigma^2 - 1.5y} & , y < 0 \\ \sigma & , y \geq 0 \end{cases} \tag{3}$$

$$f(x,y) = \frac{1}{\sigma_1 \sqrt{2\pi}} \exp\left(-\frac{x^2}{2\sigma_1^2}\right) \tag{4}$$

$$\lambda_1 = \frac{\lambda}{s} \tag{5}$$

$$g(y,s) = \frac{\lambda_1}{2} \exp\left(\frac{\lambda_1(\lambda_1\sigma^2 + 2y)}{2}\right) \mathrm{erfc}\left(\frac{\lambda_1\sigma^2 + y}{\sqrt{2}\sigma}\right) \tag{6}$$

Parameters, $\sigma$ and $\lambda$ represent the plume-spread and decay-rate of $NO_2$, with $\tau = 1/\lambda$ being the decay time or life time. The parameters $\sigma_1$ and $\lambda_1$ shown in Eqs. 3 and 5 represent the adjusted form of a plume upwind of the source ($\sigma_1$), and the decay-rate divided by the wind-speed ($\lambda_1$). Each observation $j$ can then be described by the sum of the enhancements of all
sources $i$ forming, Eq. 7.

$$Column_{NO2j}(\psi_j, \theta_j, \mathbf{s_j}) = \sum_i a_i f(x_{i,j}, y_{i,j}) g(y_{i,j}, s_j) + B_{i,j} \tag{7}$$

The emission rate of each source $i$ can then be calculated by dividing the emission enhancement $a_i$ by the decay-rate $\tau$; $E = a/\tau = a\lambda$.

In this work, a grid with a resolution of 0.1° x 0.1° is used to describe the emission, covering the full domain of Germany with
2° of padding added to the edges to reduce any edge effects (Dammers et al., 2022). The resolution is chosen as a compromise between computational burden, the limitations of the instrument, the level of detail required and conditioning of the linear system in equation 1.

The lifetime of $NO_x$ depends on both the chemical decay rate and loss to surfaces (dry deposition). Within our domain of interest, the chemical decay will be the dominant factor. Commonly used lifetimes in literature are typically based on either
modelled lifetimes or derived lifetimes from (satellite) observed plumes. Modelled lifetimes are commonly estimated via the availability of OH and production thereof (often including radiation) (Valin et al., 2013; Lorente et al., 2019). Several studies have explored this route before and either estimate the availability of OH by some basic assumptions on production, or by using modelled OH fields (with the drawback of a potential bias within the simulated concentrations). Either route is possible and

estimates for the effective lifetimes end up around 2-5 hours for spring and summertime values (Valin et al., 2013; Lorente et al., 2019). Outer estimates for wintertime lifetimes are 12-24 hours (Shah et al., 2020). Alternatively, lifetimes can be derived from tagging emitted molecules and tracking these within the model domain (Curier et al., 2014). The study reported that for a region representative of Germany (Benelux), approximately 50% of the modelled satellite signal (Ozone Monitoring Instrument, OMI, (Levelt et al., 2018)) results from $NO_x$ emissions in the 3 h prior to OMI overpass.". Assuming a relatively constant source this translates to a lifetime of about 4 hours (at column level, and assuming a basic mass balance). Several other studies report on effective lifetimes derived from fits to observed plumes from cities and large industrial areas. Using the EMG plume functions the studies derived lifetimes between 2-5 hours based on the decay downwind of major sources worldwide(Beirle et al., 2011; de Foy et al., 2015; Goldberg et al., 2021; Lange et al., 2022; Fioletov et al., 2022) with a recent study by Fioletov et al. (2022) giving a value of 3.3 hours representative for larger emissions within the US and Canada (2018-2022).

Following the modelled and observed lifetimes we assume a mean lifetime of 4 hours ± 1 hour to account for local and seasonal variations. A potential point of concern remains how representative the lifetime is for the whole year. Most of the estimates are biased towards spring, summer and autumn as there are typically more observations available within these months. To correct for the representativity bias a seasonal variation factor (1.11) will be included (explained in next section), but additionally by choosing a value of 4.0 hours we remain on the high end of the lifetime estimates. The standard deviation of ±1 hour ensure that common values within 3-5 hours remain within the uncertainty range. Furthermore, Fioletov et al. (2022) also notes that while lifetime has a large impact on the emission estimates, relative changes do not have a major impact when comparing individual years to one another. They point out that 1h deviations from the 3.3 hour mean only changed the emission estimates between years by about 1%.

The plume-spread can be seen as a combination of the diffusion, satellite footprint size, and the spatial size of the sources (McLinden et al. (2022)). Taking into account the effective TROPOMI foot-print as well as the added diffusion we use a value of 7 km for the plume-spread (similar plume spreads are used in Griffin et al. (2021b) and Fioletov et al. (2022)). A dampening factor is added to the linear system in Equation 1, forming Eq. 8 to reduce oscillation effects within the solution. The resulting sparse linear system can be solved efficiently with the *Scipy sparse.linalg.lsqr* package (Paige and Saunders, 1982; Virtanen et al., 2020).

$$\begin{bmatrix} \mathbf{A} \\ \gamma\mathbf{C} \end{bmatrix} \mathbf{x} = \begin{bmatrix} \mathbf{B} \\ \mathbf{0} \end{bmatrix} \tag{8}$$

Satellite observations of short lived species are only representative of emissions near the overpass time. A correction factor should be applied to the satellite-based estimated emissions to account for the diurnal variability. To account for this we can use a basic box-model to approximate the mass over time and apply a posterior correction. Assuming a mass $m(t)$, and a constant lifetime ($\tau = 1/lifetime$) and $E$ the emission at time $t$, the mass can be calculated with,

$$m(t) = m(t-1)e^{-\tau} + E(t) \tag{9}$$

This equation is applied to the domain-wide emissions that are injected into the domain for a whole year, including a few days of spin-up, and averaged and normalised for a selection of expected lifetimes. For the temporal distribution we use the average $NO_x$ emission profile for all $NO_x$ sources within the German domain as used in the LOTOS-EUROS model (Manders et al., 2017). A lifetime of about 4 hours and an overpass time of around 13:00 LST results in a correction factor of 1.24, meaning that the estimated emissions can be expected to be overestimated by around 24%.

Depending on the source location and time of year this value is expected to vary due to variations in the temporal emission profile. However, as actual measurements of diurnal cycles of nitrogen-dioxide emissions are rare and only exist for larger power plants, only the variability in the model emissions can be used to create a regional adjustment parameter. Surface concentration observations should in turn be used to analyse and optimize the modelled diurnal emission profiles for individual sectors. To calculate the viability of such a regional factor the adjustment parameter was calculated for each cell. The standard

deviation of the regional parameters is around $\sim 0.05$. Therefore, to reduce complexity, the value of 1.24 is assumed for the entire domain. A similar parameter is derived to account for the seasonal variability of the emissions in combination with the variable availability of TROPOMI-$NO_2$ observations passing the data quality filters. The correction parameter is calculated as the weighted mean of the number of observations per month and the mean correction factor for each month. Using this approach, a value of 1.05 is found. Combined with the diurnal parameter this gives a factor of approximately 1.30.

TROPOMI is only capable of observing $NO_2$. Therefore, an additional correction is needed to account for the NO mass. The $NO_x$ to $NO_2$ concentration ratio depends on the local chemistry, influenced by ozone concentration, photolysis frequency of $NO_2$ (solar zenith angle and cloud cover dependent), and the rate constant of $NO+O_3$ reaction (temperature) with values commonly falling within the 1.2-1.5 range for polluted regions (Beirle et al., 2011, 2019, 2021; Lange et al., 2022). In this study we apply the $1.32\pm0.26$ factor as used by Beirle et al. (2019) and include the standard deviation of 0.26 (20%) further

into the uncertainty budget account for the variations.

### 2.2.2    Method uncertainties

Based on the methods and choice of parameters described in the previous sections a summary can be made of the total expected uncertainty of our method. An overview of the uncertainty parameters with a short summary of the chosen parameter values and impact on the emissions is given in table 1.

One of the major parameters of uncertainty is the TROPOMI-$NO_2$ data product. As stated earlier, the current TROPOMI-$NO_2$ product overestimates concentrations in background/low emission regions ($\pm$a few %) while having a negative bias in source regions: -35%, up to -50% in extreme cases (Verhoelst et al., 2021), which according to van Geffen et al. (2022) adds up to a potential mean bias of around -32% to -23%. Assuming -23% for a larger industrialized region such as Germany, we end up with an underestimation of the emissions by a factor of -30%. Locally these values can decrease further (high-emission zones)

or increase (low-emission zones, up to a positive percentage). The main cause for the bias can be found in the inaccuracies of the air mass factor (AMF) which come from uncertainties in the underlying modelled concentration fields and missing variations in the stratospheric $NO_2$ concentrationsvan Geffen et al. (2022). Local variations due to errors in the AMF cannot be corrected for without the use of a chemistry transport model (CTM) and lead to under or overestimation of emissions in high

source and background regions. A recent approach using the modelled CAMS-Europe profiles (Douros et al., 2023) shows that the large negative bias can be resolved with the help of higher resolution a-priori profiles. Beside this systematic uncertainty the VCDs will also have a random uncertainty (of up to 30-50% for individual observations). Due to the large number of observations used to constrain each source, the impact of those uncertainties are expected to be minor. Furthermore, there is the detection limit of the TROPOMI instrument, which limits the ability to detect smaller sources. The study by Beirle et al. (2019) gives a limit of about 0.11kg/s, based on the divergence method. An emission source of 0.11kg/s equals about 3.5 kt $NO_x$ per year. This is based on a peak fit which typically has a radius of 25km, which roughly gives us a 2500km$^2$ area, which when divided the detection limit over the area, results in an detection limit of around 1.4 tonnes km$^{-2}$. To summarize the total expected uncertainty in the emissions due to the TROPOMI product will add up to around -30%.

The second major parameter with a large uncertainty is the choice of lifetime. An underestimation of the chemical losses could lead to an overall overestimation of the emissions, and vice versa an overestimation of the lifetime can lead to an underestimation of the emissions. A doubling of the lifetime roughly halves the emissions, which shows the importance of the parameter. Lifetimes as stated are location-dependent and to more accurately estimate will require further detailed plume and chemistry (model) studies. Examples of recent studies (Beirle et al., 2011, 2019, 2021; Lange et al., 2022) give an indication of the typical ranges for the $NO_x$ (chemical) lifetime and give a range of 2-5 hours, with the study by Lange et al. (2022) giving a value of 3-5 hours representative of the Germany domain. The $4\pm1$ hour results result in an -33 to +20% under/overestimation of the emissions.

The $NO_x$ to $NO_2$ ratio can also have local variations, which affect the total emissions. At source level the majority $NO_x$ is emitted as NO, which can rapidly turn into $NO_2$ after which an equilibrium is reached, the speed of which depends on the availability of $O_3$. Beirle et al. (2021) recently gave a modelled estimate of the ratio, which was very close to the factor 1.32($\pm$20%) given in his original study, with values moving towards 1.0 for industrial areas just north of the equator while values tended towards higher ranges (1.6) for less industrialized and high-latitude regions.

Next up, there is the influence of the wind-speed and direction, for which we assume an uncertainty of up to 1ms$^{-1}$ (McLinden et al., 2022) in both the U and V wind field parameters, leading to the realistic situation of a higher uncertainty in direction at low wind speeds. The effect translates into an uncertainty of around 15-20% for average conditions over Germany (based on the matched wind-fields) which matches earlier uncertainty estimates by Griffin et al. (2021a); McLinden et al. (2022).

Finally, the diurnal and seasonal variations show some variations in the order of a few percent (<5%). Note that a fixed parameter was determined for the whole German domain, but locally the diurnal correction factor can be lower/larger for the more continuous/strongly varying emissions. For example, in the case of power plants, which run more continuously than road transport, this can result into a negative bias for the emissions.

Taken together, these error terms result into a Germany averaged error range between -50 to +35% for the Gaussian plume method. The low error estimate corresponds to source regions where the low bias of the TROPOMI VCDs, effectively low biasing the emissions, are counteracted by the potentially high bias in the emissions of the $NO_x$:$NO_2$ ratio and effective lifetimes. Both values should be seen as conservative estimates, which would occur in the unlikely case that the inaccuracy in $NO_x$:$NO_2$ ratio, lifetime, AMF and wind-fields all nudge the estimate in the same direction for all locations in the domain of

interest. In reality not all errors point in a similar direction, like the product bias pointing in opposite directions for background
and source regions.

| Parameter | Summary | Impact on final emissions(%) |
|---|---|---|
| TROPOMI: AMF/other bias | -23% mean bias | -30% |
| TROPOMI: Noise | 30-50% for individual observations depending on VCD range | Minor, large number of observations reduces uncertainty. |
| TROPOMI: Detection limit | 3.5 kt $NO_x$ for isolated individual sources | $\pm1.4$ tonnes km$^{-2}$ |
| **Total: TROPOMI** | | **-30%** |
| Lifetime | 4 hours$\pm25$% | -33% to +20% |
| $NO_x$:$NO_2$ ratio | 1.32$\pm0.26$ | $\pm20$% (a factor of 1.41 gives an increase of +7%) |
| Wind Fields | $\pm1$m s$^{-1}$ | $\pm15$-20% |
| Diurnal & Seasonal emission cycles | 1.3$\pm0.05$ | $\pm5$% |
| **Total uncertainty** | | -50% to +35% |

**Table 1.** Summary of uncertainty parameters for emission estimates

### 2.2.3 Sector specific emissions

A direct sector based attribution of emissions is not feasible using the satellite data only. Therefore, additional data needs
to be taken into account to attempt to estimate a potential sectoral attribution of the emission. We used the GNFR/CLRTAP
sector outputs to create a spatial index filter for the emission data. The GNFR data is used as a basis and summed and re-
370 gridded for all the NFR classes, to match the 0.1°x0.1° grid used in this study. A Gaussian filter (*scipy.ndimage*,Virtanen et al.
(2020)) is applied to the data with a sigma of 1 grid cell. The posteriori smoothing is only there to bridge the limitations
of the method and instrument. The spatial limit to resolve 2 sources of similar size depends on the effective lifetime, the
pixel size and meteorological factors such as typical diffusion etc. Of these the pixel size and lifetime are dominant at the
TROPOMI pixel limit (5.5x3.5km$^2$). The pixel size combined with diffusion gives us a typical plume width of around 7 km
(e.g. $\sigma^2=\sigma^2_{plume}+\sigma^2_{pixel}+\sigma^2_{source}$). This value varies depending on typical size of a source but most sources of $NO_2$ are limited
in size (except for large mines, very large cities etc). Based on McLinden et al. (2022) a plume-width of 7 km combined with
a lifetime of 4 hours gives an effective resolvability limit of 15-20km, which for 0.1°x0.1° source cells (e.g. $\sim$10x10km$^2$) ex-
plains the choice for a sigma of 1 grid-cell. More smoothing can produce better results, but also reduces the observable details.
SSIM should be seen more as a metric to judge comparability, and not accuracy of the emissions, as the inventory emissions
are not perfect either. The resulting masks are divided by the total emissions of all sectors to derive each sector's fraction of all
emissions (emission fraction), and shown in the supplement Fig. A1 for the non smoothed version and Fig. A2 for the Gaussian

smoothed version. For further sectoral emissions analysis only locations with an emission fraction above 50 % are selected, the resulting mask is shown in Fig. A3.

## 3   Results

### 3.1   Inter-comparison with the emission inventory

For a comparison with the gridded inventory data, we used the 2019 data from the satellite derived emissions and the respective $NO_x$ data from the GNFR inventory (CLRTAP, 2021). Figure 3 gives a visual comparison of the 2019 data sets. Both sets were compared using the Structural Similarity Index Measure (SSIM) (Wang et al., 2004) for a quantitative comparison of the images. The SSIM operator is a metric, which was developed to evaluate the image quality of video frames . It uses a window based comparison analysis to track subtle differences between two images so that the spatial structure of both images is also taken into account when calculating the SSIM score. This way similarity and dissimilarity between two 2D data sets may be quantified with the SSIM score in a way which assesses image similarity in a more human-vision-based mode. Since its introduction SSIM has become a standard comparison operator for computing the similarity between 2D data sets and is now also available in standard open-source data analysis packages such as scikit-image (van der Walt et al., 2014).

The resulting SSIM analysis for the 2019 GNFR and TROPOMI derived emission data shows a SSIM score value of 0.6 between both data sets. However, to consider the different approaches of both data sets and to harmonize effects of different baseline resolution, a Gaussian filter is applied to both sets of data that compensates the effect of the larger point spread function (PSF) of the sensor. If both images are Gaussian filtered and compared, the resultant SSIM score is 0.79 and now closer to a score of 1.0, which would depict spatial structure identity between the two sets of data. This illustrates that the spatial structure of both data sets show a high similarity and space-borne derived emissions from the method presented here capture similar large emission sources such as major cities, road networks and industrial areas as the GNFR data set.

### 3.2   Multi-year emissions

The satellite derived emissions for the individual years between 2019-2021 are shown in Figure 4. The rightmost plot shows the emissions as part of the GNFR inventory. A comparison between the satellite derived emissions of individual years and the inventory emissions for 2019 are shown in Fig. 5. Figure 6 shows the change in the spatial satellite derived emission distribution of the largest sources in Germany between 2019 and 2020 respectively 2021. As the gridded inventory emissions are only available for 2019, we can only compare the individual years to that years inventory emissions.

Fig. 4 shows that the spatial data from the 2019 space-borne emission estimates have elevated $NO_x$ emission values of around 5-7t/km$^2$ that seem to coincide with the major motorway network in Germany. Most notably is the enhancement observed near motorway A2 westwards from Berlin, via Magdeburg and Hannover towards the Rhine-Ruhr region. Another high emission region seems to be co-located with the motorway A1 from the Ruhr area of North-Rhine-Westphalia towards Bremen and Hamburg. The motorway ring and spider-alike road networks and settlements fanning out from around Berlin seem

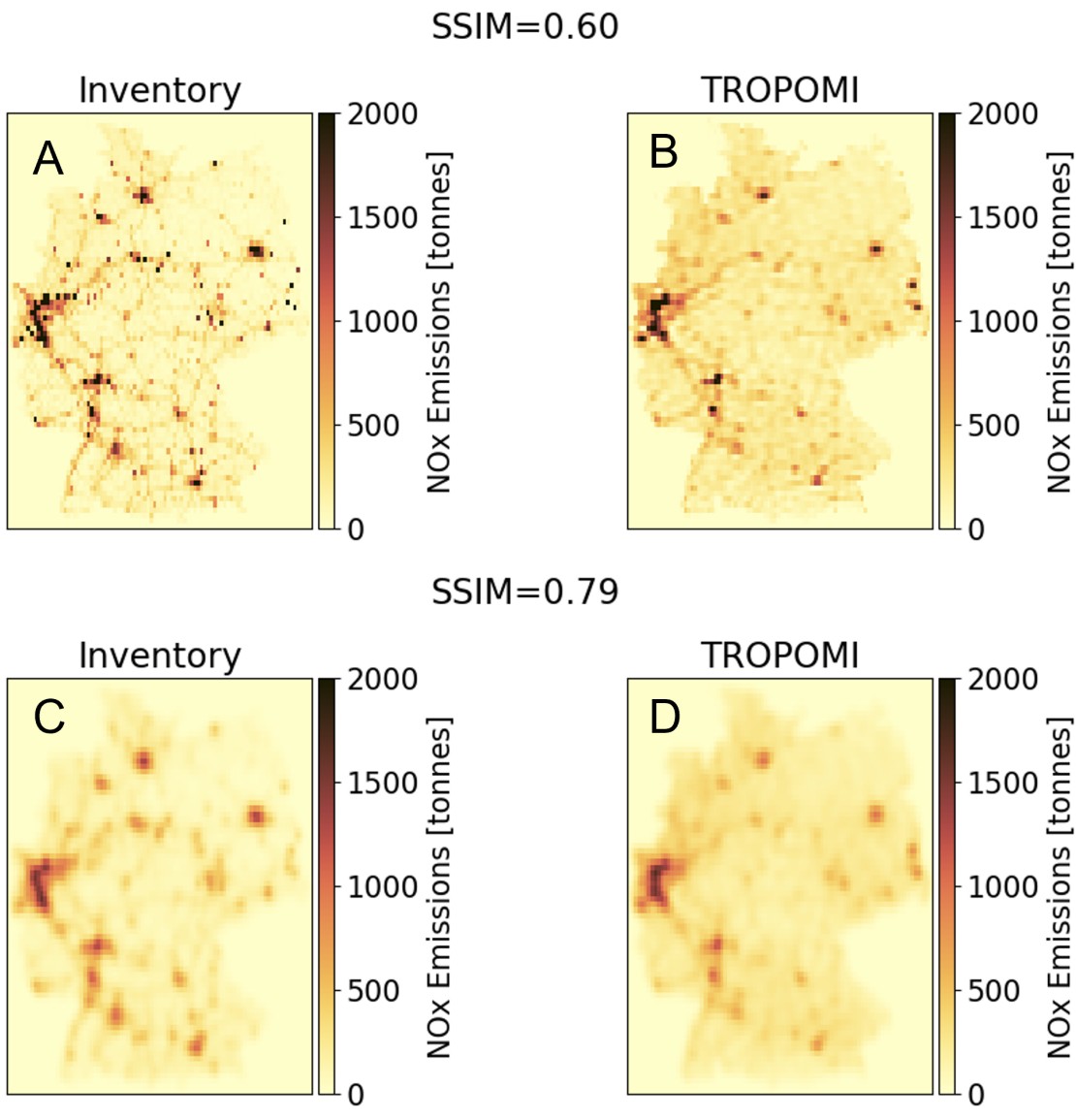

**Figure 3.** The Structural Similarity Index Measure SSIM of 0.6 was calculated between the gridded inventory data (top-left panel, A) and the emissions derived with the TROPOMI data for 2019 (top-right panel, B). Please note that the details in the (image-) data structure (location of major road networks and urban areas) are very similar between both sets of data. This is highlighted by a SSIM score of 0.6, which quantifies as the similarity between the data as highly significant. If the data is Gaussian filtered effects of spatially sharper GNFR data (panel C), compared to TROPOMI data (panel D) are compensated and yield a SSIM score of 0.79

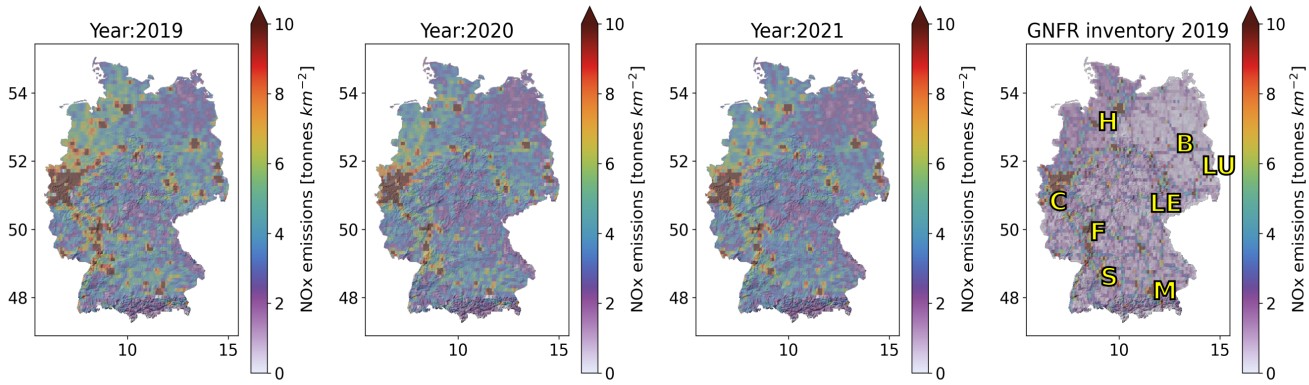

**Figure 4.** From left to right; NO$_x$ satellite derived yearly emissions for 2019-2021, and the GNFR inventory emissions of 2019. Rightmost figure H: Hamburg B: Berlin, C: Cologne, LU: Lusatia, LE: Leipzig, M: Munich, S: Stuttgart, F: Frankfurt.

to be likewise visible in 2019. Caution has to be taking with attributing emission to road networks since other high emissions sources, e.g. industrial cites, tend to be located in close vicinity to major traffic arteries.

While the TROPOMI instrument represents a huge step in the capabilities to spot individual emission sources, there are still limits to the spatial resolvability. The top row of Fig. 5 shows a direct comparison while the bottom row shows the same results but now with the application of the Gaussian filter, as previously used in Fig.3. The main difference between these two rows are the large positive/negative swings around the more localized emissions and/or major point source like emitters such as power plants, visible in the top row without the Gaussian-filter. Such variations are, however, not observed around emitters with large

spatial footprints such as cities. This is an excellent showcase of the limit of the method and TROPOMI's spatial resolution. Through the size of the satellite pixel's footprint and the misrepresentations of the wind fields (i.e. artefacts) there is an actual limit to the overall spatial resolvability of individual sources. This limit was reported by McLinden et al. (2022) to be around 5-10km for TROPOMI, which matches well to the size of the source grid used here. The 0.1°x0.1° spatial resolution used in this study is thus at the limit of the methods capabilities to constrain individual neighbouring sources and some smearing is

thus expected around the strongest point like sources. The Gaussian (smearing) filter can be used as a first order correction, which results in the lower row of plots. Compared to the inventory emissions of 2019, the Figs. 5 D, E and F show similar patterns between the years, with strong negative differences observed around the major sources, while the background regions (i.e., regions with emissions below 2 tonnes km$^2$) show a consistent positive difference of around 0-1 tonnes km$^2$. There are several potential causes for these systematic patterns which will be evaluated in the discussion section.

Outside of the systematic patterns there are several variations visible between the years. The year 2020 shows a noticeable drop in NO$_x$ emissions around industrial, cities, and highways (Fig. 6). The largest reduction of NO$_x$ emissions between 2019 and 2020 are in the industrial areas in the Rhine Ruhr region and the upper Rhine area. The rise in emissions from 2020 to 2021 in the TROPOMI data in Fig. 4 and Fig. 6 is most noticeable in the larger urban areas, which is most notable in Fig. 6.

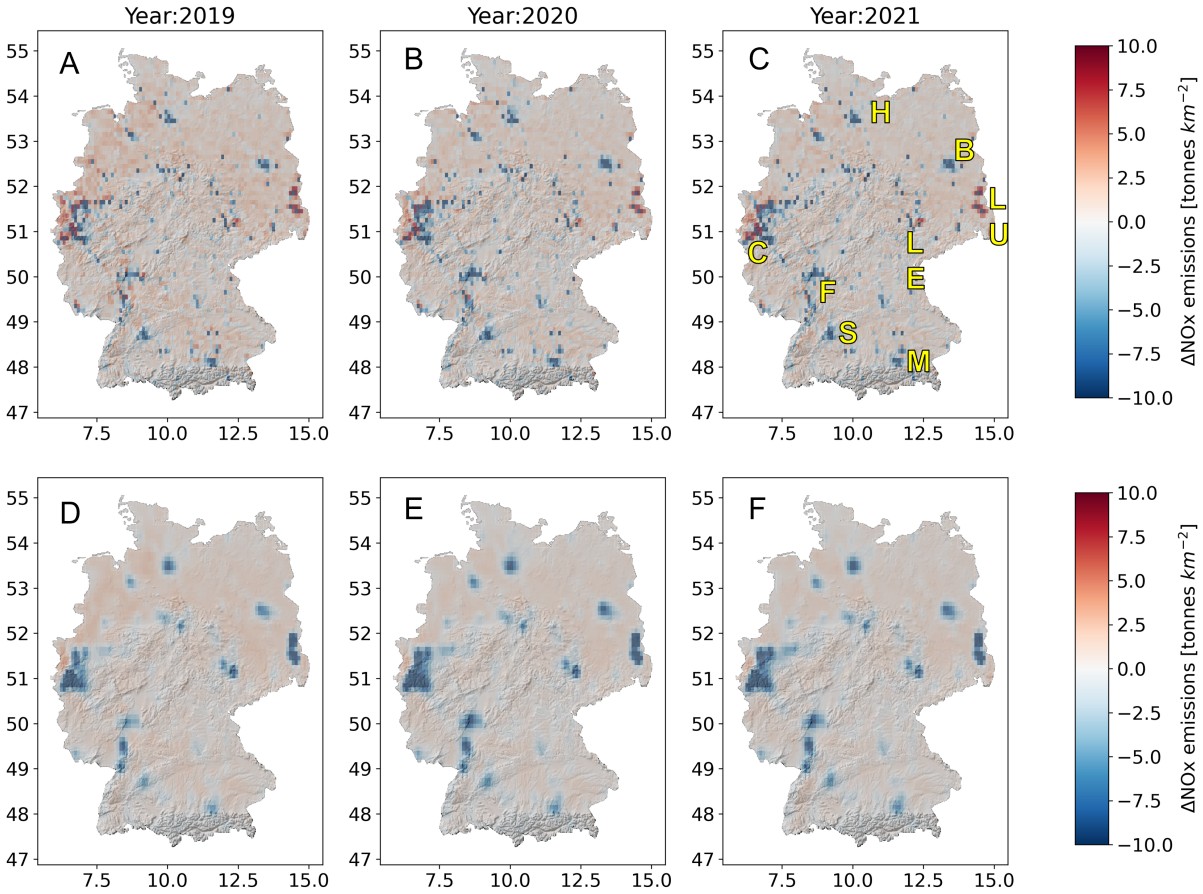

**Figure 5.** Difference between the satellite derived and inventory emissions (2019) for years 2019-2021 (A-C). The red values indicate a higher value for the satellite derived emissions compared to the inventory emissions. Top row shows the original difference between both emission sets, the bottom row shows the same sets but now with the Gaussian filter applied to both sets before subtracting the 2019 inventory emissions (D-F). The letters in the figure indicate the following; H: Hamburg B: Berlin, C: Cologne, LU: Lusatia, LE: Leipzig, M: Munich, S: Stuttgart, F: Frankfurt.

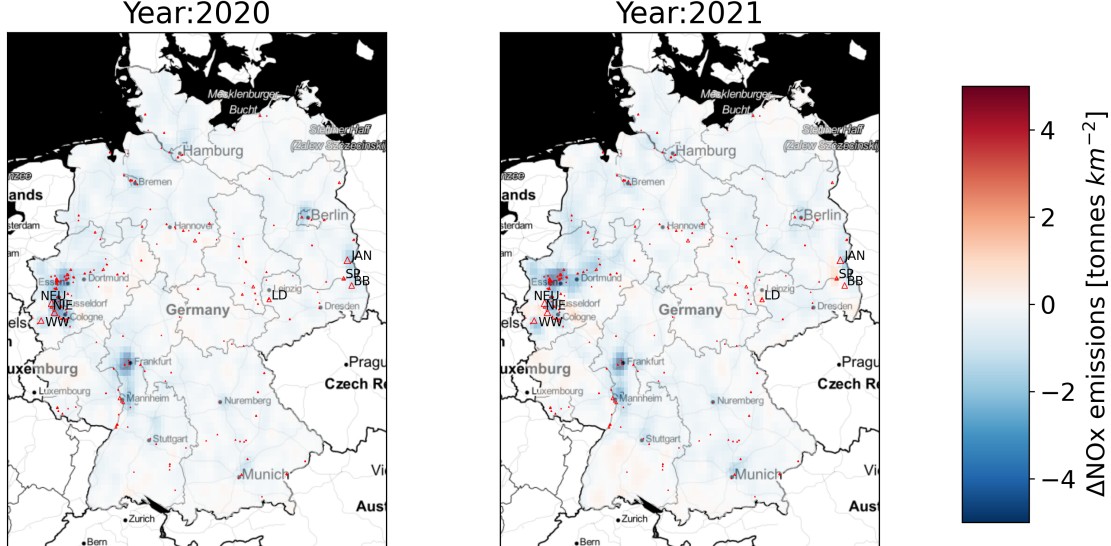

**Figure 6.** Difference between 2020 and 2019, respectively 2021 and 2019 satellite derived emissions. A Gaussian filter has been applied to both derived emission sets. The red dots indicate the locations of the largest NO$_x$ emitters within Germany, with the size of the dots a reflection of the individual source strength. The red triangles indicate the larger power plants, with the letter combinations indicating the names of the power plants; NEU (Neurath), NIE (Niederaussem), WW (Weisweiler), LD (Lippendorf), JAN (Janschwalde), SP (Schwarze Pump), and BB (Boxberg).

However, the 2021 NO$_x$ emissions are still lower than in 2019, for example in the industrial centers of the Rhine-Ruhr region
(note the red dots indicating the major industrial emitters) and further south along the Rhine. Only the A1 motorway (the line of emissions between the major emissions clusters at C and H) is still clearly visible in the 2020 and 2021 emission estimates (Fig. 4), whilst the data from other settlement and road networks (e.g., around Berlin) is much less obvious than in the 2019 emission estimates. This is also visible in Fig. 6 in the area with roads leading away from Berlin, where the difference between the 2021 and 2019 estimated emissions still show negative difference of the order 0-1 tonnes km$^2$.
Two of the most prominently visible changes (2019-2020) shown in Fig. 6 are the industrial "Ruhr" region, the largest and oldest industrial core of Germany in the westernmost part of the country, and the area of Lusatia in the eastern part of the country with a large-scale lignite mining industry to supply coal fired power plants. These two areas are shown as detailed maps in Fig. 7. Compared to 2019 the emissions have dropped substantially in 2020 and 2021 (up to <-5t/km$^2$). The power generation in Germany has seen an increase in the usage of coal fired power stations for power generation in 2021 compared to
the COVID-19 year 2020, as reported by the DESTASIS in its press briefing (link) stating that coal had been the most important source of electricity generation in Germany in 2021. This can be seen in Fig. 7, where there is an increase near one of the large

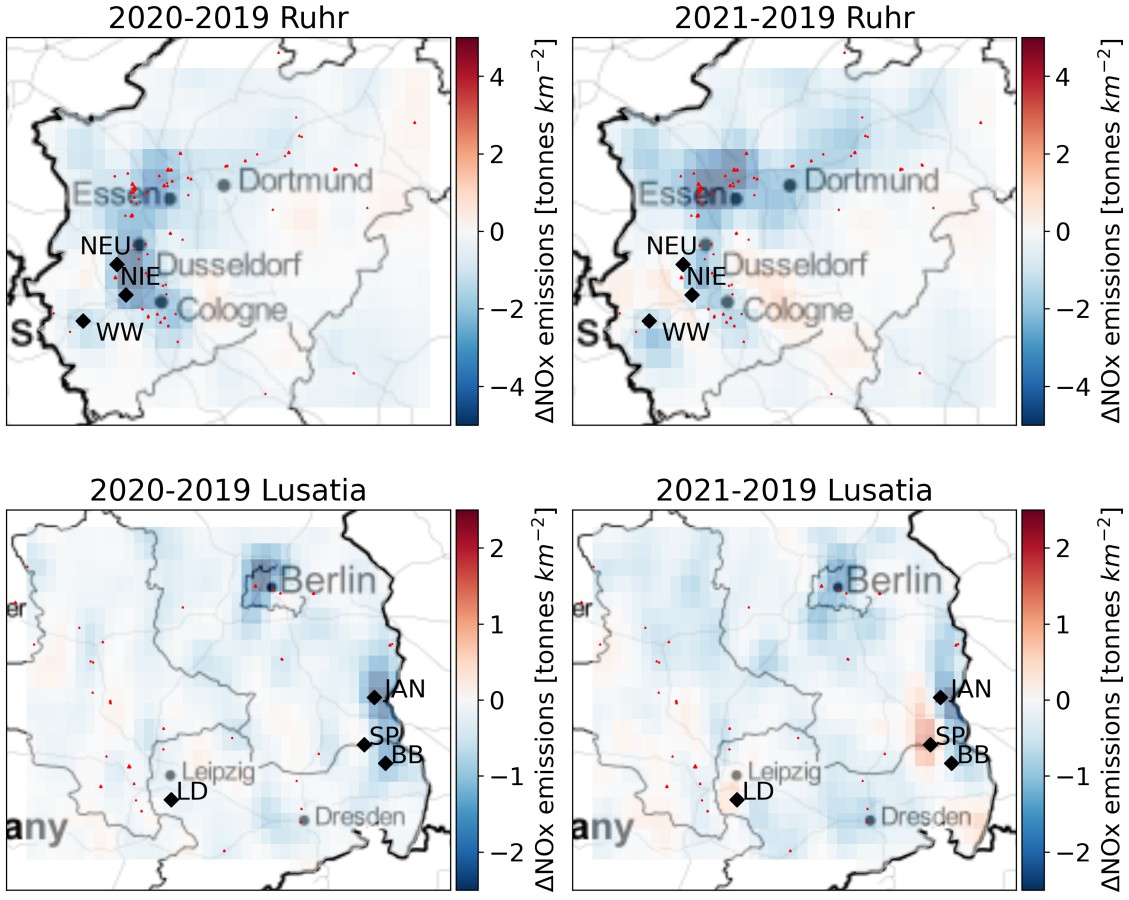

**Figure 7.** Difference between 2020 and 2019, respectively 2021 and 2019 satellite derived emissions. Upper row depicting the industrial "Ruhr" region, whilst the lower 2 panels show Lusatia at the eastern border of Germany. A Gaussian filter has been applied to all data sets prior to subtraction. The red dots indicate the locations of the largest $NO_x$ emitters within Germany, with the size of the dots a reflection of the individual source strength. The black diamonds indicate the larger power plants, with the letter combinations indicating the names of the power plants; NEU (Neurath), NIE (Niederaussem), WW (Weisweiler), LD (Lippendorf), JAN (Janschwalde), SP (Schwarze Pump), and BB (Boxberg).

emission centers right at the eastern border of Germany. The Schwarze Pump (SP) and Lippendorf (LD) power plants even show an increase in emissions compared to 2019. Meanwhile the emissions from the Janschwalde (JAN) power plant show a strong reduction in 2020 that continues into 2021, which was expected as the power plant reduced its operation capacity as planned (Vattenfall, 2015; EPH, 2022). The three large power plants in the west show similar patterns with the Neurath (NEU) and Niederaussem (NIE) plants showing a strong decrease in 2020 rebounding upwards moving into 2021. The Weisweiler power plant reduces into 2020 while reducing further into 2021. This drop can be explained by two potential causes. Firstly, a planned reduction in operation capacity and secondly by the flooding from exceptional rainfall in mid July 2021, which also affected the nearby lignite mining pits (RWE statement / Link to news item).

## 3.3 Sector-specific emissions

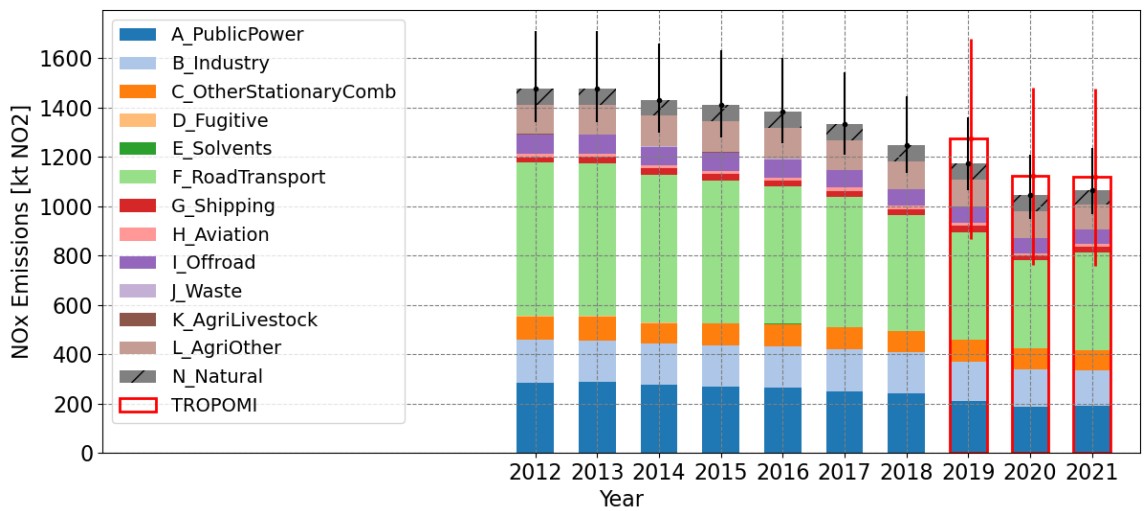

**Figure 8.** Emission changes over the years as reported by the national inventory of Germany and observed by the TROPOMI instrument. Black error bars indicate the uncertainties on the inventory emissions while the red error bars show the uncertainty in the satellite derived emissions. Note the slight rise in the reported emissions of 2021, compared to the year 2020 due to COVID-19 lockdown measures.

An aggregated version of the spatially distributed results is shown in the bar plot of Fig.8, in which the country wide fitted emissions are compared to the country-wide sector specific emission totals. Note, that we added the 60 kt $NO_x$ from natural soil emissions and 5 kt $NO_x$ from lightning emissions to the N_Natural class. These emissions were not included in the previous spatial plots. In line with the previously discussed results, both the satellite and inventory emissions show a large drop from 2019 to 2020, of comparable size. The slight increase in the projected inventory emissions from 2020-2021 is however not matched by a change in the satellite emissions.

Emissions sources that have a strong spatio-temporal imprint on TROPOMI data should show independent patterns for regions where the sources cause the majority of emissions. To find out what type of source is causing this mismatch, we make use of the sectoral masks (e.g., fig. A3) to derive sector specific patterns from the spatio-temporal data from the satellite derived and inventory emission data.

Only 5 sectors (public power, industry, road transport, shipping and agriculture other than livestock) have locations which are dominated (e.g., above 50% of the total emissions) by a single emission sector, of which the public power sector has the largest emissions in a single location while the road transport emissions are more spread out over roads and pastures throughout the country. Note, that the public power, shipping and industrial emissions cover a very limited area with only public power showing very high emissions. Figure 9 shows the sector specific emissions as indexed by the 2019 emissions for the public power, industry, road transport and shipping sources. Based on earlier projections, and trends over the previous years, 2021 inventory emissions are expected to be just over 90% of the 2019 emissions. The emissions related to power generation have bounced back after the pre-COVID-19 levels even though the Janschwalde power plant in the east reduced its operation capacity as planned (Vattenfall, 2015; EPH, 2022). The emission in 2021 showed a recovery to 93% of the pre-covid estimates. Further resurgences are to be expected (for 2022) by the plans of the reactivation of old coal fired power plants in the wake of the European energy crisis and the potential fears of a blackout in Germany. While road transport emissions were expected to show a recovery this is not matched by patterns in the satellite derived emissions. The slow recovery can potentially be explained by the reduced number of kilometers by trucks (vehicles with a weight above >3500kg) which is almost down by 10% in 2021 compared to 2019 (KBA, 2022). Shipping emissions have continued their decline with no sign of recovery. While this reduction was expected based on past trends, the cause can be found in the global shipping crisis and disrupted supply chains.

## 4 Discussion

As the results showed, the captured spatial variability within satellite derived emissions are very similar to those in the analyzed inventory emissions. The values for the $NO_x$ emissions retrieved from the TROPOMI observations diverge on average by 75-100 kt $NO_x$ (<10%) from the emissions reported for Germany (Fig. 8). There are some variations observed between the years but the difference between both emission estimates fall within the uncertainty range of both emission totals. The uncertainties of both emission estimates are quite large compared to the yearly variations, which hampers stronger conclusions on the quality of the inventory and satellite based estimates. We can however discuss the various causes of uncertainty and how these can be reduced. The uncertainty range of the reported inventory emissions is estimated to fall between -9.2 and +15.8% (see https://iir.umweltbundesamt.de/2022/), which translates to about -100 to +180 kt $NO_x$ in 2019 for the inventory. Note that this range does not include potentially missed sources, such as stronger than expected natural emissions (e.g., Soil emissions) and any of the MEMO items.

Besides the above discussed items it should be noted that the emissions from road transport are required to be based on the fuels sold approach. Additionally, this this approach does not account for all the emissions which occur in Germany from

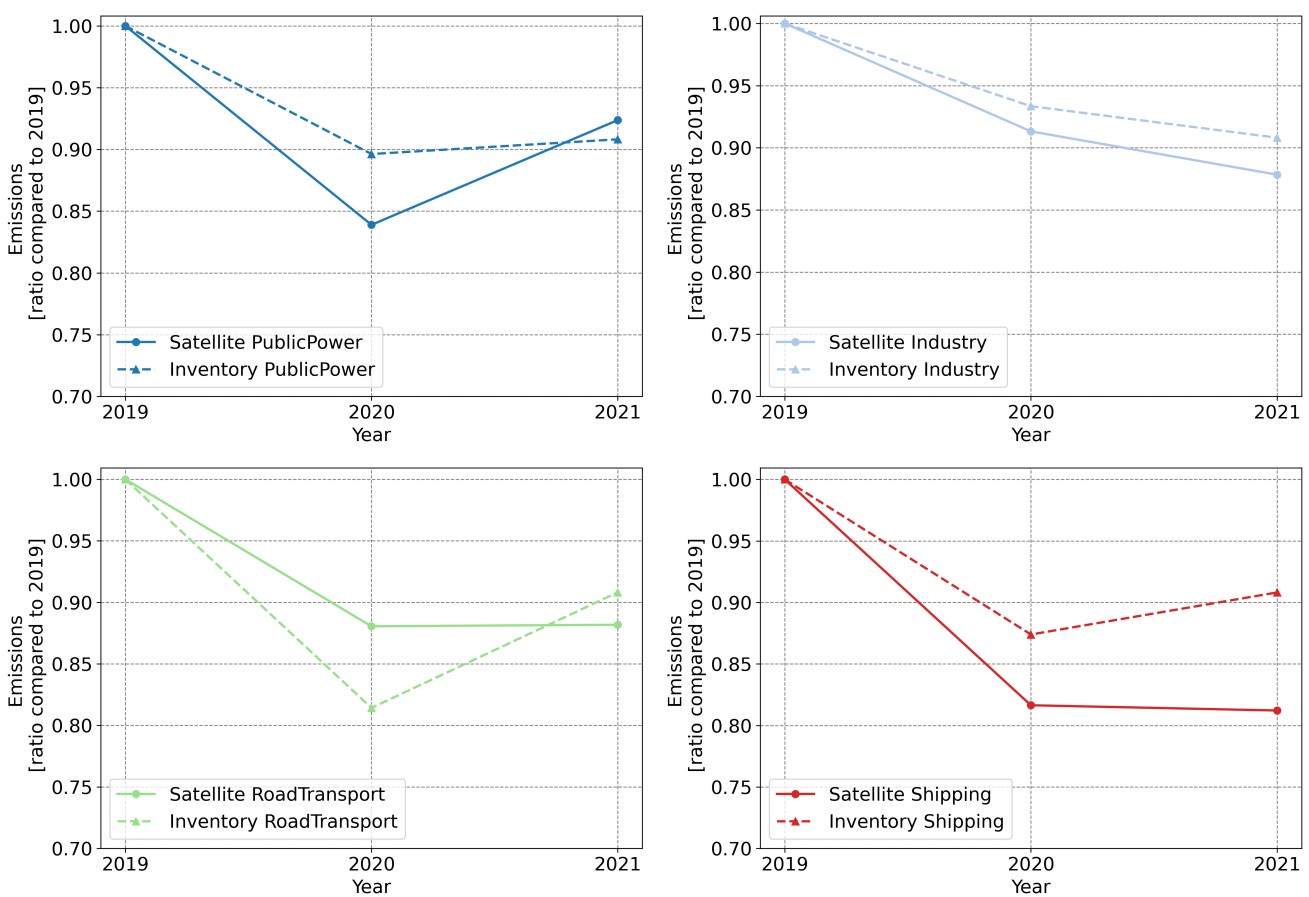

**Figure 9.** Satellite derived and inventory emissions for each source sector as indexed by the 2019 emissions. A clear decline is visible for most sectors for 2020 w.r.t to 2019. Dotted lines indicate the inventory emissions, the solid line the satellite derived emissions.

vehicles which were fueled abroad and are driving in Germany (this might constitute an underestimation in the inventory). On the other hand the emissions from foreign vehicles (for instance from the Netherlands) which bought their fuel in Germany and were not driving in Germany are in this fuel sold approach allocated to Germany (this might constitute an overestimation of the German emissions). However, it is not known how much emissions are associated with these cross border phenomena's for Germany. Data from The Netherlands show this might be a significant difference; the $NO_x$ emission based on fuel used is approximately 5.5 % less than the emissions based on fuel sold as reported in the GNFR total. However, as fuel prices in Belgium and Germany are cheaper than in The Netherlands, Dutch drivers fuel frequently in those countries thus the Dutch case represents the higher end of the difference between fuel sold and fuel used approach, only surpassed by Luxembourg with one of the lowest fuel prices in Europe.

Another source of uncertainty are the emissions near the border regions. Emissions within the first 10 to 20km outside of the border can be expected to be smeared out in the satellite derived emissions due to the limited resolvability of the instrument and methodology. The stronger the source, the better the resolvability. So for the larger sources 10 km can be assumed. Making a loop around the German borders there are a few areas of interest. Starting at the border of the Netherlands and moving clockwise on both sides of the border, there are several larger sources, such as the Weisweiler power plant in Eschweiler, the Dolna Odra power station in Poland, and several power plants near the border in the Czech Republic, but also several smaller and larger cities. By taking a polygon that is 10 km wider and narrower in shape than the existing German borders the smeared emissions near the borders can be approximated. Based on the European CAMS-REG v5.1 inventory (emissions 2018, based on the in 2020 reported emissions(Kuenen et al., 2022)), we find that around 120 kt $NO_x$ of the German emissions take place within Germany and within 10 km of the borders, and around 75 kt $NO_x$ just outside of Germany within 10 km of the border. Assuming that at most half of the full amount of these emissions smear out past the border, the smeared loss in emissions is about 22kt on the total emissions. This should be seen as an upper limit. Furthermore, of these emissions a large majority takes place in the western part of Germany, where the most common wind direction is wind coming from the west. In effect it can be expected that the smearing of those emissions will be reduced further.

A more probable source for the 100 kt $NO_x$ mismatch however can be found in the satellite derived estimates. As stated in section 2.2.2 the TROPOMI based emission estimates can have an uncertainty in the range of 35-50% translating to about +-400 kt $NO_x$. A more complete error analysis based on simulated observations with controlled conditions and a subsequent Monte-Carlo analysis of error propagation could give a more accurate estimate, but falls outside of the scope of this study. A study by Dammers et al. (2022) however did perform such an analysis. While using a very similar set of input parameters, the study derived a mean uncertainty between 15-20% which increases when close to large mountains. Two important differences between this manuscript and that study (Dammers et al., 2022), are the uncertainties and bias in the satellite product (which only shows a minor negative bias) and the lack of a $NO_x:NO_2$ ratio. Without both parameters the 15-20% uncertainty would translate into an uncertainty of around $\pm150$-200 kt $NO_x$.

For the regional emission mismatches, of the uncertainties studied in Dammers et al. (2022) and in this study, only the lifetime, the satellite product bias, and $NO_x:NO_2$ parameters can have a large enough systematic effect on the estimated emissions to explain the observed differences. Additionally, one could argue that the wind fields around larger hills and mountains can

have a systematic effect. However, throughout our region of interest most of the mismatches (Fig. 5) are observed away from the main mountainous regions. The negative bias observed around the major emitters (up to and over 50%) can be explained by both the product bias, an increase of $NO_x:NO_2$ ratios near the source (Lange et al., 2022; Griffin et al., 2021a), and a mismatch in the $NO_x$ lifetime. The product bias by itself can be expected to cause an underestimation of at least -20% (van Geffen et al., 2022). A higher $NO_x:NO_2$ ratio of 1.5 at the upper end of the literature values (Lange et al., 2022; Beirle et al., 2021; Griffin et al., 2021a) would result in an additional underestimation of about -15%. The lifetime values reported in literature show a more random variation. Assuming the 3.3 hour estimate from Fioletov et al. (2022) is more accurate for emission zones, this would add an additional 20% low bias to our estimates. Taken all together these values add up to an underestimation of about -45%, which is close to the observed difference. The positive difference observed away from the major emitters, and especially in regions with intensive use of arable land[6], could potentially hint at underestimation of soil emissions throughout Germany (Fig. 5). As discussed in section 2.1.1, soil emissions show a large range within the literature, with strong variations due to the availability of nitrogen, soil type, humidity and temperature. Some variations are observed between the years which could reflect changes in any of these parameters. The detection limit of the instrument does not seem like a likely candidate as it would result in low bias, similarly the slight high bias of the product cannot explain the larger differences observed in the northwest. The other two parameters that can cause a systematic offset, the $NO_x:NO_2$ ratios and lifetime, also do not seem to be a logical suspect. The $NO_x:NO_2$ ratio can at most cause a few percent of a positive bias (i.e. a ratio of 1.25 would only result in a few percent difference) while the lifetime would need to double or triple to explain the difference observed in the north-west.

The year to year variations in the TROPOMI-$NO_2$ derived emissions are of the order of a few to ten % (Fig 8). While the estimated errors of individual years are larger than those variations, most error components will stay consistent between the years. A similar conclusion was made by Fioletov et al. (2022) who performed various experiments to test the impact of a common offset, in lifetime and plume-width, on the emission estimates of several years. In our case the consistency of the TROPOMI product version ensures that negative bias in the TROPOMI product can be expected to stay stable between the years over the high VCD regions while staying slightly positive over background regions. The only terms that are expected to slightly change are the $NO_x:NO_2$ ratio, the effective lifetime, and changes in wind-patterns. The changes in wind-patterns will only matter for regions in the border regions as misinterpretation of the wind-fields will typically results in the wrongful attribution of emissions within Germany. This leaves the $NO_x:NO_2$ and effective lifetime as the main source of uncertainty, both related to the timing of emissions and the chemistry. A potential method to constrain this effect is performing a CTM run over the same period but with fixed yearly emissions over the whole period. The emission estimate methodology of this study can then be used to estimate the emissions of the individual years and thus derive the influence of changing chemistry and meteorology. This however falls outside of the scope of this study.

---

[6]https://www.eea.europa.eu/data-and-maps/figures/agricultural-land-use-intensity-1

## 5 Conclusions and Outlook

This work has shown that TROPOMI can be used as a verification tool for emission inventories, even for those inventory compilers, which are unfamiliar with remote sensing data. Emission inventory compilers may monitor near-real-time trends in $NO_x$ emissions with the tool via top-down space-borne data without the need to wait for the completion of the statistical data required for the classic statistical "bottom-up" approach for the calculation of emissions. This is of particular importance for the quantification of unforeseen events such as the outbreak of the COVID-19 pandemic, which has been shown in this paper by comparing the 2019 emission data to the COVID-19 (2020) and post-COVID-19 year (2021). Individual sectors are, however, difficult to assess given the low spatial resolution of TROPOMI. However, if we look at single large contributors to emissions such as the public power sector shown in Fig. 9 it is possible to track the rebound in emissions after the corona year 2020. Which has been due to the increased usage of coal fired power plants for power generation in 2021 compared to 2020. Similar trends and changes in $NO_2$ concentration may now be assessed by the emission inventory community worldwide as they are now able to compare their countries results to others using the here presented fully transparent methods. This has previously not been possible in a convenient way for inventory compilers. As at least comprehensive data science knowledge is required to access and query other data products e.g. from the ECMWF atmospheric data storage (ADS[7]), the web tool is complemented by the source code offer, which specifically invites other developers to extend the space-borne emissions code-base and web-tool by their own contributions.

Space-borne data from TROPOMI and other satellites contain valuable information that can be used as a verification tool for emission inventories. $NO_x$ retrievals from space-borne sensors such as OMI and TROPOMI can be used to monitor the quite dramatically decreasing evolution of $NO_x$ emissions over the years with new emission estimation methods such as (Fioletov et al., 2017). Although, sub-sector and facility related data still is difficult to assess, the data still delivers valuable insight into the coarser spatial distribution of emission clusters, such as the chemical industry parks around Halle and Leipzig or large coal fired power stations in the east of Germany. This may help to monitor emission reductions directly of these large industrial clusters. This satellite-based emission estimates, based on a single consistent methodology applied to several countries, can be used to verify the compliance towards meeting the air pollution reduction targets throughout the whole of the European Union, which ensures a maximum of transparency for all stakeholders. This ultimately values the principals of the European Green Deal initiative[8], which tries to leverage new technology for a sustainable EU.

With the presented space-emissions tool other emission inventory compilers without remote sensing expertise are encouraged to employ space-emissions for verification of their inventories. This would make the space-emissions tool a critical building block of emission compliance reporting thanks to the Copernicus Sentinel data set (i.e., TROPOMI) that are provided by ESA. We are looking forward to the feedback from the emission inventory community and their results using the online and offline tools. The here developed methodology and online (and offline) tool was initially focused on $NO_x$ emission estimates from TROPOMI observations. In the future the incorporation of OMI data would extend the time series to the year 2005, which is of great importance for the verification of a more complete time-series of the inventory. The coarser resolution of

---

[7]https://ads.atmosphere.copernicus.eu

[8]https://commission.europa.eu/strategy-and-policy/priorities-2019-2024/european-green-deal_en

the OMI observations (being coarser that the 0.1º x 0.1º resolution used in this analysis) will however lead to a less detailed emission map. The additions of other pollutants should also be envisioned for future work under the reservation that the respective method is applicable to the selected pollutant. In the near future the geostationary Sentinel-4 satellite is scheduled for launch and will provide hourly data on tropospheric constituents over Europe. This will allow tooling such as those used in this manuscript to explore additional functionality such as the measurement of time profiles and might allow for emission estimates on a weekly of even daily basis and provide information on the diurnal emission cycle. While the methodology was only applied on a yearly basis in this study, TROPOMI has enough spatio-temporal coverage to move to seasonal or monthly estimates, potentially trading the spatial resolution of the emission fields for an increase in temporal resolvability.

Future improvements to the methodology should focus on updating the AMF with the help of higher resolution modelled fields, the addition of a location dependent lifetime (for example based on concentration of $NO_2$, $O_3$ and OH), the addition of local $NO_x$:$NO_2$ ratios and local corrections for diurnal and seasonal cycles) which all three would make sense from a physical perspective and form the largest uncertainty in the method outside satellite bias. Some of these improvements require simulated model fields, of which some are available in the form of (open-access) CAMS ensemble runs. Other required variables such as temperature, UV radiation, precipitation and humidity, which would be used for adjusted lifetimes, are also available at the various ECMWF data storage. These quantities and/or estimates can be downloaded with the ERA download tool and already make a relatively easy improvement to the lifetime estimates and thereby reduce the overall uncertainty of those terms.

*Code and data availability.* An offline version of the emission code is available at https://github.com/UBA-DE-Emissionsituation/space-emissions. All code used to produce further results figures etc, can be provided on request. The TROPOMI L2 data product versions (OFFL / PAL) can be accessed through the ESA Sentinel-5P data hub (https://s5phub.copernicus.eu) and the PAL data-portal (https://data-portal.s5p-pal.com/). The emission inventory data-sets can be accessed via https://iir.umweltbundesamt.de/2022/, and the GNFR/NFR data sets via Link to NFR and Link to GNFR.

## Appendix A: Additional figures

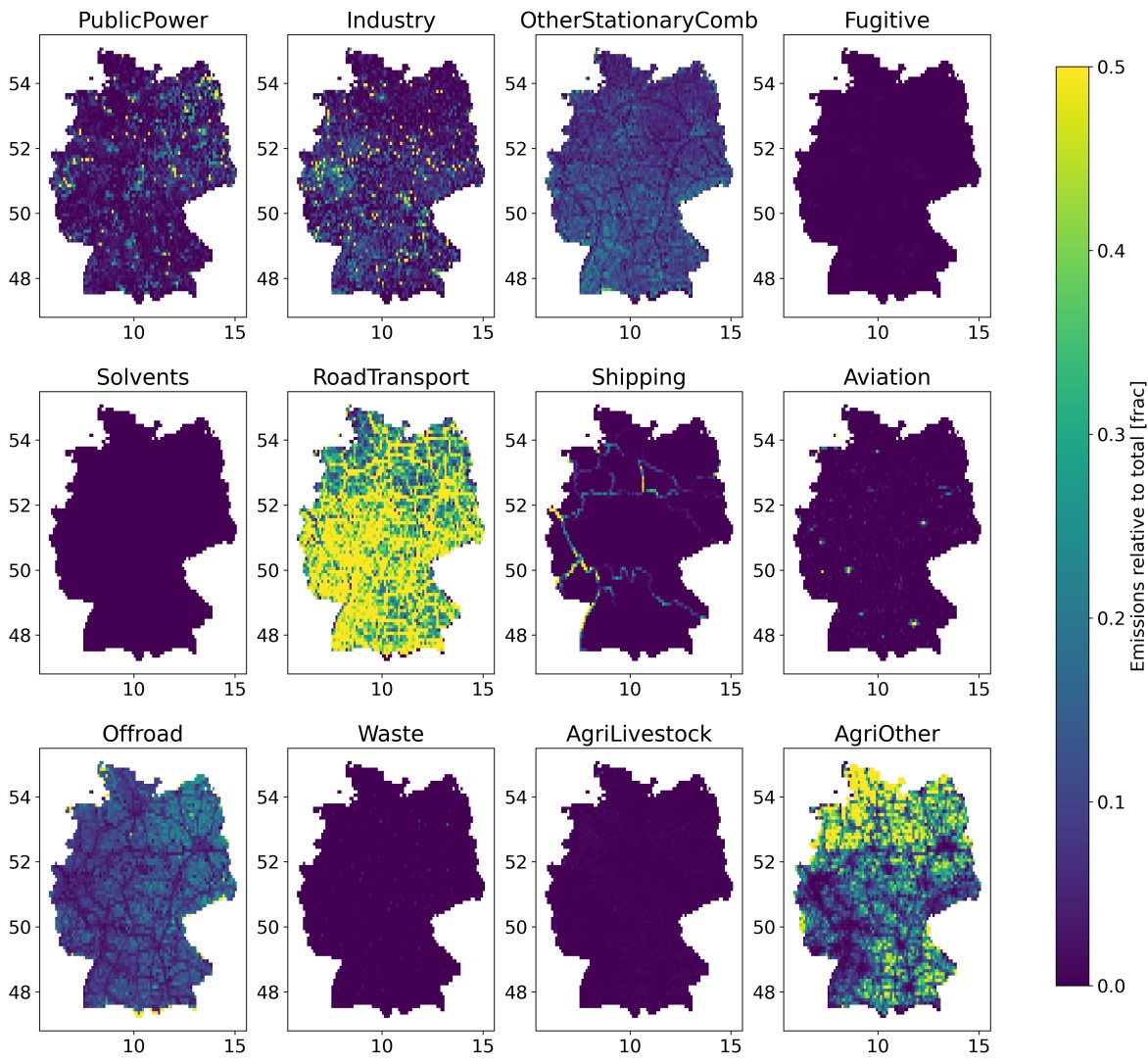

**Figure A1.** Fraction of NO$_x$ emissions emitted by each emission sector for each grid cell within the German domain. Yellow indicates locations with emissions dominated (>50%) by an individual source sector. The displayed data is based on gridded GNFR inventory emissions of 2019.

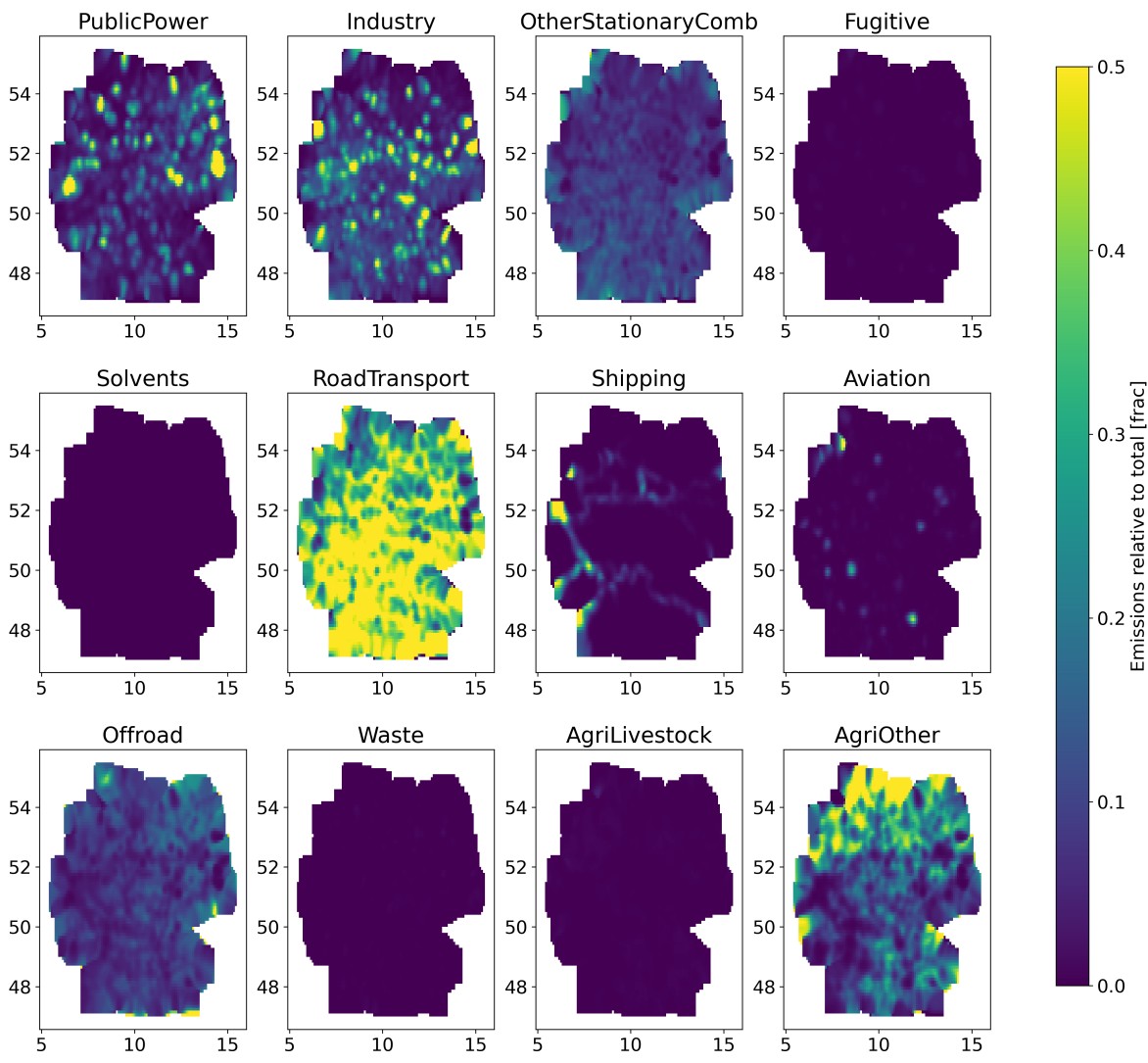

**Figure A2.** Fraction of NO$_x$ emissions emitted by each emission sector for each grid cell within the German domain, smoothed with gaussian. The displayed data is based on gridded GNFR inventory emissions of 2019.

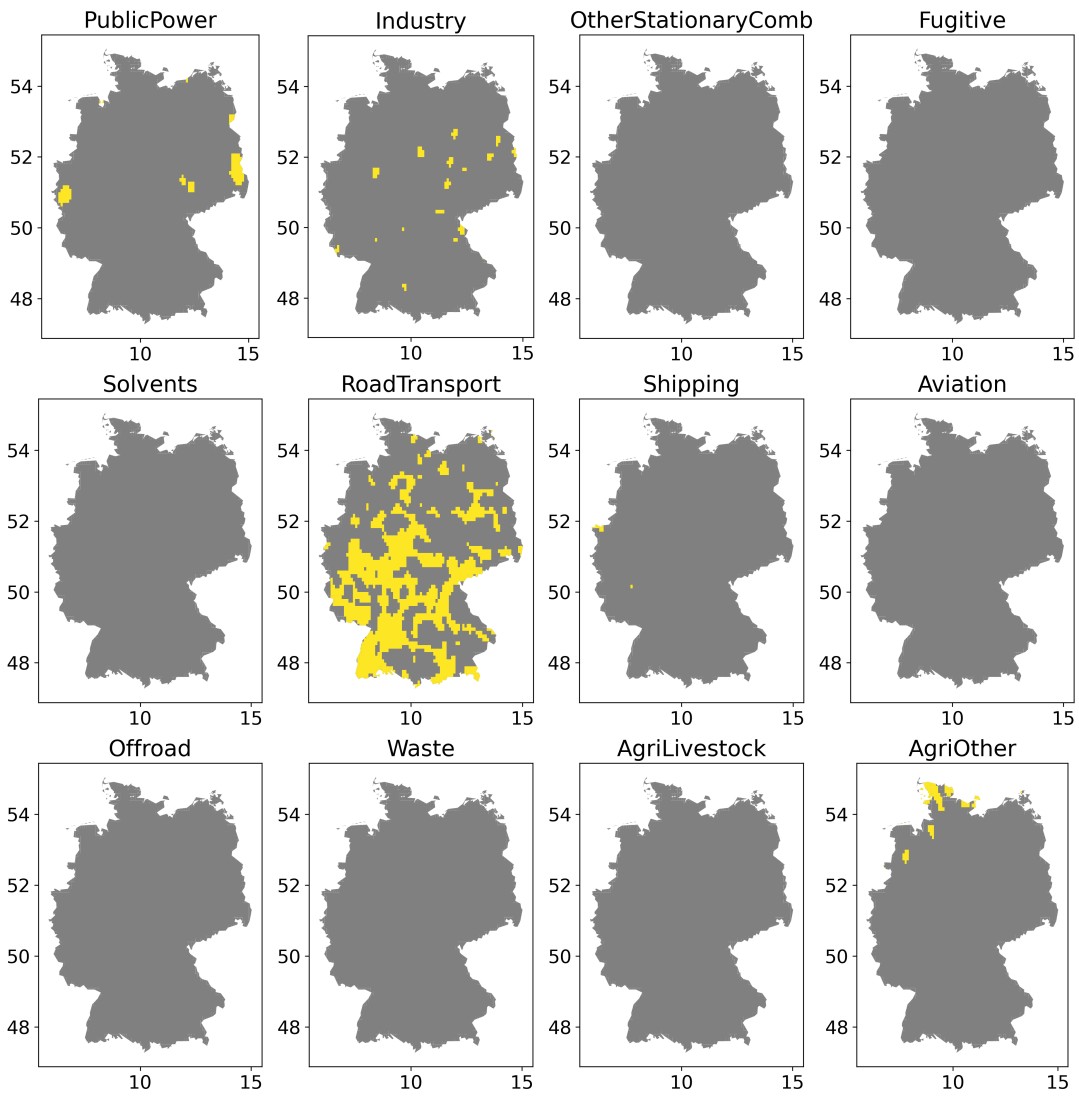

**Figure A3.** Emission source locations selected to produce sectoral trends. The produced masks are based on the results shown in Fig. A2, for all locations with an emission fraction above 50%. used to distinguish different source sectors in the emissions derived from satellite data.

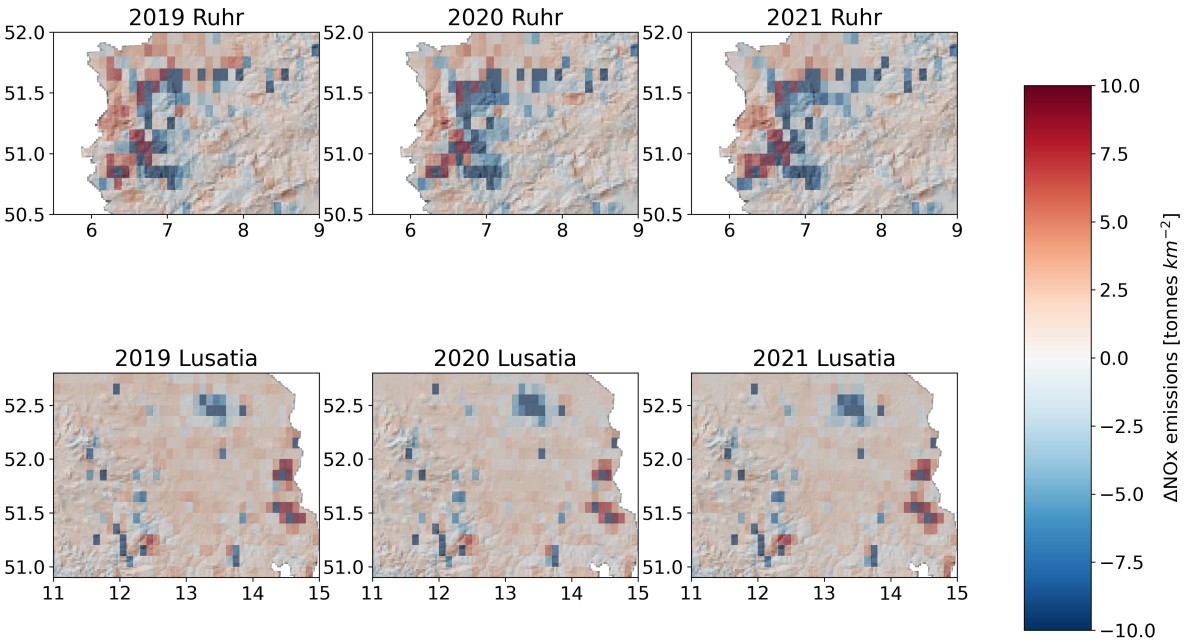

**Figure A4.** Difference between the satellite derived and inventory emissions (2019) for years 2019-2021 over two zoom regions. The red values indicate a higher value for the satellite derived emissions compared to the inventory emissions. Upper row depicting the industrial "Ruhr" region, whilst the lower 3 panels show Lusatia at the eastern border of Germany.

*Author contributions.* Enrico Dammers devised and implemented the methods presented in this paper and carried out the data analysis and writing of this publication together with Janot Tokaya, who made critical contributions to the data stream handling of the method (retrieval of the correct scenes, and CAMS meteorology data). Renske Timmermans coordinated the scientific work from the TNO side and gave critical input to the scientific work with respect to the atmospheric chemistry of $NO_x$. Christian Mielke interpreted the spatial data of the algorithm runs and wrote the parts of the publication that deal with the emission inventory relevant topics. Kevin Hausmann designed and implemented the webtool and the level-0 emission estimation on the Code-DE Platform and provided the critical scientific environment for the success of this project. Debora Griffin and Chris McLinden helped devise the methods presented in this paper and gave critical input to the manuscript. Henk Eskes provided information on the satellite product. Finally, all authors discussed the results and reviewed the manuscript.

*Competing interests.* The authors declare that there are no competing interests present in carrying out the work presented in this publication. The work has been financed by the Umweltbundesamt in the Leitplankenproject (FKZ: 3720515010).

*Acknowledgements.* We acknowledge the hard work done by KNMI, ESA, the team behind the PAL data portal, and the TROPOMI teams for making TROPOMI a success and providing easy access to the $NO_2$ data. ECMWF ERA5 data Hersbach et al. (2020) was downloaded from the Copernicus Climate Change Service (C3S) Climate Data Store (https://cds.climate.copernicus.eu/cdsapp#!/dataset/reanalysis-era5-pressure-levels?tab=overview). We thank Stefan Feigenspan for providing the 2019 GRETA data and for sharing his knowledge on emission gridding.

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
