# Peer review of "Can TROPOMI-NO2 satellite data be used to track the drop and resurgence of $NO_x$ emissions within Germany between 2019 - 2021 using the multi-source plume method (MSPM)?"

_Geoscientific Model Development, 2022_

## Author Comment (AC1)

We would like to thank reviewer 1 for his/her constructive comments and suggestions.

**General points:**

1. **The manuscript needs to be thoroughly checked for the logic and for the English grammar. Many statements are unsupported or confusing. Many commas are misplaced, making sentences difficult to read and understand. Some sentences make no sense by themselves, but do make sense if joined to the next sentence. Many citations use parentheses incorrectly. Subscripts are sometimes missed.**

   Response: The manuscript has been edited in detail by a native English speaker. Furthermore, we tried to make all requested changes. Instead of listing all the edits here, we point towards the author's tracked changes document in the next submission.

2. **The authors assume an NO2 lifetime of 4 hours when deriving emissions, but shouldn't this lifetime impact the footprint on the sources when comparing to the satellite data? With 5 m/s winds for example 4 hours of advection transports the NOx ca. 72 km, but the Gaussian smoothing used to produce Fig. 3 uses a σ of 1 grid cell, ca. 10km. Would more smoothing produce better SSIMs and more accurate inversion of the emissions?**

   The footprints of a source is indeed linked to the lifetime of the emissions. But the lifetime is taken into account while creating the source-receptor relations, the Ax=B linear system, and thus the inversion. The inversion approach is the leading driver in producing accurate emission totals. The posteriori smoothing is only there to bridge the limitations of the method and instrument. The spatial limit to resolve 2 sources of similar size depends on the effective lifetime, the pixel size and meteorological factors such as typical diffusion etc. Of these the pixel size and lifetime are dominant at our ~5.5x3.5km2 pixel limit. The pixel size combined with diffusion gives us a typical plume width of around 7 km (e.g. $\sigma^2_{plume} = \sigma^2_{plume} + \sigma^2_{pixel} + \sigma^2_{source\_size}$). This value varies depending on typical size of a source but most sources of $NO_2$ are limited in size (except for large mines etc, and one could argue a city can be one larger source).  Based on McLinden et al., 2023 (to be submitted but showing an example figure here) a plume-width 7 km combined with a lifetime of 4 hours gives an effective resolvability limit of 15-20km, which for 0.1x0.1degrees source cells (e.g. ~10x10km2) explains our choice for a sigma of 1 grid-cell. More smoothing can produce better results, but also reduces the observable details. SSIM should be seen more as a metric to judge

comparability, and not accuracy of the emissions, as the inventory emissions are not perfect either.

[Figure]

**Figure E2.** Minimum distance required to distinguish between two point sources as a function of plume width and effective lifetime. The line plots to the left and below show this quantity for $\sigma = 11$ km and $\tau = 3$ hours, respectively, and are cross-sections indicated by the white dashed line.

3. **Related to this, the TROPOMI-derived emission uncertainty is stated to be 30–40% (p21, L428), but it isn't clear where these numbers come from. The text on p21 discusses many sources of uncertainty (including the lifetime issue), but how were these combined to make 30–40% and the error bars (whatever they are) in Fig. 8?**

   Response: The discussion on the uncertainties (lines 430-479) has been moved and extended to form its own section (2.2.2, from line 290 onward) which can be read in the updated version of the manuscript. Additionally a table (1) has been added to summarize the individual uncertainties/errors. Some further explanation has been given for the individual error terms, linking back to earlier sections in the manuscript where needed. The discussion section has also been shortened to account for the moved section.

4. **p2, L39–49, and also p20, L403 onwards. I missed a discussion of the known problems of real-world emissions. For example, as cited in Oikonomakis et al. (2018), several studies showed a significant discrepancy (a factor of 2–4) in the NOx emissions from light-duty diesel vehicles between two driving cycles, and Anenberg et al. (2017) and found similar issues for heavy duty vehicles, indicating inadequacy of the testing procedures to capture real-world emissions. The diesel-gate**

**scandal was also a good example of the limitations of emissions reporting (Jonson et al., 2017).**

Response: The reviewer nicely points out why there is a need for more validation and verification of the emission inventories using independent data streams. As the reviewer notes there can certainly be further "unknown unknowns" in the emission inventories. Independent verification, for example based on satellite observations, could more rapidly trace and reveal discrepancies. While the diesel-gate scandal is the obvious example, it is complicated to point out additional "unknown unknowns" without moving too much into speculation territory. We added a few sentences to the discussion on the importance of independent verification to trace and reveal potential "unknown unknowns".

Added to introduction: Furthermore, accurate methods that allow for independent verification, can potentially be used to trace and reveal significant discrepancies in the current emission inventories. An example of a discrepancy in the past would be the diesel-gate scandal which was a good example of the limitations of emissions reporting (Jonson et al., 2017).

**Other points:**

**• p1, L12, Explain 100kt as percentage, so that the reader knows if this is a large or small number.**

Response: Changed 100kt to 75-100kt (<10\%) at L12.

**• p1, L14, Add a year here so that the reader knows when this recovery was 'projected'.**

Response: Changed line to "The recent projections for the inventory emissions of 2021 pointed to a recovery of the 2021 emissions towards pre-COVID19 levels this increase was not observed."

**• p1, L17. It sounds odd that satellites help faster "projections", since that term is usually reserved for future values. Re-phrase.**

Response: Changed sentence to "This again illustrates the value of having a consistent satellite based methodology for faster emission estimates to guide and check the conventional emission inventory reporting.".

**• p1, L18. Change meet to meets, or method to methods. Check such things throughout the manuscript.**

Response: As mentioned above, edited the document. Changed to "meets".

**• p2, L2. Add also a more recent reference than Crutzen 1970.**

Response: Changed to Seinfeld and Pandis 2006.

**• (nit-picking I know): Say 'largely' primary. Some NOx is produced by lightning, and NO2 is mainly produced from ozone reactions with primary NO.**

Response: Added "for the most part" L23 P2.

**• p2, L27. The units should be formatted correctly, not in italics. Also, be consistent. The unit on L28 has a space between g and m, whereas on L27 it doesn't.**

Response: Formatted throughout document added a space at L27.

**• p2, L30. Give reference for the statement about acidification and eutrophication.**

Response: Added reference: Galloway et al., 2003.

**• p2, L36. There is no such thing as the 'Geneva Convention ...', in this context at least. The authors mean the Air Convention or CLRTAP equivalents, e.g. https://unece.org/environment-policy/air. 'Nations' is also not an appropriate reference; maybe 'UN-ECE' or similar.**

Response: Correct, realizing the drawbacks of the Overleaf editor. Changed accordingly.

**• p2, L44. The word projections confused me here and elsewhere. Usually the term is used for future scenarios, e.g. for 2030 or 2050. Here I think the authors just mean emission estimates.**

Response: Changed to "emission estimates".

**• p3, L60. Give info on this 'unprecedented horizontal resolution', or refer to appropriate section for details.**

Response: changed to "with its unprecedented spatial resolution of $3.5 \times 5.5 \text{ km}^2$"

**• p3, L65. Refer to appropriate section for details of code and availability.**

Response: Removed "latest draft", links to code and availability are located in the code and data availability sections.

**• p3, L71. Tell the reader where this is 'described further'.**

Response: Removed line, added url to https://space-emissions.net/.

**• p3, L75 (also p7). What is Umweltbundesamt (UBA) for those not familiar with German institutions?**

Response: Added German Environment Agency.

**• p3, 1st paragraph. This section is somewhat repetitive of Sect. 1, and isn't really 'Methodology and Datasets'. Some is also repeated, or better placed, in Sect. 2.1.**

Response: Moved part of the section to the introduction, removed "The national inventory data is reported through the informative inventory report (IIR): For the case of Germany it is publicly available from its original source: " as it was repeated at a later point.

**• p4, L104. Where does this 300% number come from?**

Response: The 300% comes from:
[https://iir.umweltbundesamt.de/2023/general/uncertainty_evaluation/start](https://iir.umweltbundesamt.de/2023/general/uncertainty_evaluation/start)) ;
*"Compared to other pollutants, NOx emission uncertainties are moderate. The national total has a 95% confidence interval of about -8.5% to +15.0% in 2021, which amounts to about 230kt of NOx. Interestingly, with NOx, the differences between the two approaches in uncertainty combination (EP and MC) are particularly visible. This is because of the highest contributing sector 3.D - Agricultural Soils, where emissions and uncertainties are high **(> +300%)** and, crucially, do not follow a normal distribution. Therefore, only the MC simulation, which takes the log-normal distribution of these emissions into account, correctly reflects this source, while the EP yields unrealistic high uncertainties at about 15.4% in both directions.*

**• p4, Sect. 2.1.1. Some of the text here is also more introductory material (e.g. L104 onwards) than technical description of the emission inventory.**

Response: Removed lines 104-108.

**• p4, L120. Change 'mol' to emissions (mol is a unit, not a quantity).**

Response: Changed to emissions.

**• .5, L124. NOx is a mixture of NO and NO2 , so one needs to specify the assumed molecular weight associated with your 5 kt NOx per year figure, or state as e.g. kt(N) NOx per year.**

Response: Added ($NO_2$) to line 124.

**• p5, L130 onwards. Same issue with NOx units and emission amounts.**

Response: Added line to state same usage of kt (NO2) unit throughout the document when mentioning NOx.

**• p5, L151. What is ATBD?**

Response: Added "Algorithm Theoretical Basis Document".

**• p6, L166. Say 'well correlated with ...'; the sentence was difficult to read**.

Response: Rewritten as The TROPOMI NO2 data correlate well when compared to ground-based MAX-DOAS and PANDORA instruments (Verhoelst et al., 2021) but tend to show an underestimation of the tropospheric column.

**• p6, L165–173. Various statistics are given concerning bias, but are the instruments being compared with (MAX-DOAS, PANDORA) free of bias themselves? Are some of the difference due to problems with these instruments?**

Response: The MAX-DOAS and PANDORA instruments are not completely free of bias themselves but typically have much lower uncertainties than the TROPOMI-NO2 product as stated in Verhoelst et al., 2021.

**• p7, L205. Mangled 'from in the naive'?**

Response: Broken link, fixed accordingly L205, changed to "uses the TEMIS monthly L3 data product available at https://www.temis.nl/.".

**• p10. The 'Column' term in Eqns. (2) and (7) looks very ugly. Use a symbol, as is done for all other terms. In any case, shouldn't this be VCD?**

Response: Changed to V, removed NO2/source indicator j.

**• p10, equations (3)–(6). Give in order of usage, thus σ1 before f (x, y), λ1 before g(y, s).**

Response: Reordered

**• p10, L255. Where is ai explained?**

Response: Its rewritten from equation 2, so stated in line 239. Made italic to make clearer.

**• p10, L258: 'Following Beirle et al. (2016) we assume a lifetime of about 4 hours(±25%)'. I can't find the terms hour or lifetime in Beirle 2016, and that paper deals mainly with the stratosphere. Why didn't you use estimates of NO2 lifetime from LOTOS-EUROS for Germany? Does the 25% estimate really capture the uncertainty here?**

Response: Note that a detailed answer to a similar question has been given in the response to Reviewer 2 (second question). To answer the specific question of this reviewer: the year stated for Beirle et al's manuscript was incorrect and should be 2019. A similar number is stated in Beirle et al's 2011 manuscript. Based on results in earlier studies (Goldberg et al., 2021, Fioletov et al., 2022, Beirle et al., 2011,Lange et al., 2022) on average the 25% estimate should hold. The results (Fig 6b/d) shown in the study by Lange et al., (2022) seem the most representative yet for our study, who give an average range of 3-5 for Paris/Northern Latitudes (49-56).

We did not use estimates of NO2 lifetime from LOTOS-EUROS as there is currently no option within the model to directly write out lifetimes. We can look at earlier studies that used a tagging approach to label emissions from individual hours. An earlier study by Curier et al., 2014 did just that to study the source sector contribution of emissions from individual hours to the OMI NO2 column at OMI overpass for several industrial regions in Europe. For the region somewhat representative of Germany (Benelux) the study states: "Approximately 50% of the modelled OMI signal results from NOx emissions in the 3 h prior to OMI overpass.". This statement holds for most the source sectors. Assuming a relatively constant source this translates to a lifetime of about 4 hours (at column level, and assuming basic mass balance).

To summarize we changed lines 258-259:

The effective lifetime of NOx depends on both the chemical decay rate and loss to surfaces (dry deposition). Within our domain of interest the chemical decay will be the dominant factor. Earlier studies using the EMG plume functions derived lifetimes between 2-5 hours based on the decay downwind of major sources worldwide \citep{Beirle_2011, deFoy_2015,Goldberg_2021, Lange_2022, Fioletov_2022}. Following those results we assume a mean lifetime of 4 hours +- 1 hour to account for local and seasonal variations.

**• p11, L287. I didn't find the factor 1.32 in Beirle et al. (2016) either.**

Response: The year stated for Beirle et al's manuscript was incorrect and should be 2019. Changed to 2019.

**• p11, L292-293: 'The gridded NFR data .... summed to the ... grid'. Does gridded data need to be gridded? CLRTAP inventories are usually gridded by NFR categories.**

Response: changed the sentence to "The GNFR data is used as a basis and summed and regridded for all the NFR classes, to match the 0.1°x0.1° grid used in this study."

**• p12, L307. Why a comma after 'way'? This is just one example of a common problem.**

Response: A native English speaker re-editted the document and made changes throughout the manuscript.

**• p12, L323. Again, somewhat sloppy. Fig. 4 doesn't say anything about previous sensors, and the text doesn't explain what the authors are thinking. If you make comparative statements, back them up.**

Response: Removed sentence.

**• p12, L330. Here I also wonder about the 4-hour footprint issue mentioned above.**

Response: Accounted for advection/diffusion in the source-receptor relations. See the more detailed explanation in the major comments.

**• p14, L345. Remove 'the before mentioned'.**

Response: Removed "the before mentioned"

**• p14, L349. How should non-Germans know where the A1 motorway is?**

Response: Added "(the line of emissions between the major emissions clusters at C and H)"

**• p15, Fig 5. Explain letters in top-right fig. Also explain whether positive values (red color) means that the satellite has more or less emission than the inventory.**

Response: "added, The letters in the figure indicate the following; H: Hamburg B: Berlin, C: Cologne, LU: Lusatia, LE: Leipzig, M: Munich, S: Stuttgart, F: Frankfurt.", also added "The red values indicate a higher value for the satellite derived emissions compared to the inventory emissions.", typically positive values, here red, indicates that x is larger than y in difference between x and y.

**• p16, Fig 6. What are the triangles? What are the various letters (NEU, WW, …)? The latter are explained in the text, but the caption should be informative.**

Response: Added The red triangles indicate the larger Power-Plants, with the letter combinations indicating the names of the powerplants; NEU (Neurath), NIE (Niederaussem), WW (Weisweiler), LD (Lippendorf), JAN (Janschwalde), SP (Schwarze Pump), and BB (Boxberg).

**• p16, L372. Why are 65 kt NOx added only at this stage?**

Response: As discussed in the manuscript the 60+5kt NO2 emissions are rough indications of the totals within the domain, but no spatially varying result could be calculated. Hence the values were only added at this stage.

**• p18, Fig. 8. Again, the caption explains too little. What are the error bars? Be explicit and say slight rise in 'reported' emissions.**

Response: added "Black error bars indicate the uncertainties on the inventory emissions while the red error bars show the uncertainty in the satellite derived emissions. Note the slight rise in the reported emissions of 2021,"

**• p18, L375. Start a new paragraph frpm 'Emissions sources¨, so that the reader knows the subject has changed.**

Response: Added break

**• p18, L380. Fig. A3 doesn't support that the Agri emissions are spread out across the country, at least not if the text is about the >50% region.**

Response: Removed agricultural emissions.

**• p18, L382. Why are 'non-agricultural sources' mentioned here? Only 3 sources are addressed, so many sources are excluded.**

Response: changed to " Public Power, Industry, Road Transport and Shipping sources"

**• p20, L395. When starting the discussions, be explicit that the reported emissions are for Germany.**

Response: rephrased to "from the emissions reported for Germany".

**• p20, L402. Why does Fig. 5 'hint' at a small and widespread source? The values seem close to zero in most areas.**

Response: While the contribution is small the value (at 2.5 tonnes/km2) is above the estimated detection limit (1.4 tonnes km2) and due to its wide-spread occurrence adds up to quite a total.

**• p20, L403. Again start a new paragraph when the subject is changing.**

Response: Added breaks throughout the discussion.

**• p21 L432. 'improved' - from what?**
Response: rewritten section on uncertainties, removed.

**• p21, L436. 'approach u' typo**

Response: Removed u.

**• p21, L437. CAMS-Europe - reference?**

Response: Added Douros et al., 2023.

**• p21, L440. 'detect' should be 'detection'**

Response: removed in rewritten section.

**• p21, L445. Mangled sentence.**

Response: removed in rewritten section.

**• p21, L445. How do you know that the 4h timescale is correct for Germany as a whole in 2019?**

Response: See earlier discussion on lifetime and responses to Reviewer 2.

**• p21, L448. Lifetimes are not only location dependent; they depend on**

**complex interactions between meteorology, chemistry and vegetation state (via deposition).**

Response: See earlier discussion on lifetime and responses to Reviewer 2.

**• p21, L453. What is meant by several %? That number seems low.**

Response: Removed in rewritten section, added % in renewed section 2.2.2.

**• p22, L472. What is '(8)'?**

Response: Added "Fig. "

**• p22, L484. The web-tool is mentoned, but I would hardly say it was 'presented'.**

Response: Removed text.

**• p23, L496. What is 'ads'? Provide a web address.**

Response: Added web address. https://ads.atmosphere.copernicus.eu/#!/home

**• p23, L498. No need to use words like 'truly' in a scientific statement.**

Response: Removed sentence.

**• p23, L502. It was discussed earlier that the resolution is not really 3.5x5.5km2, and the derived-emissions resolution are certainly not at that level.**

Response: Removed part of sentence.

**• p22-24. Again, one sinlge paragraph over more than a whole page! Break up the text into separate topics.**

Response: Added breaks throughout.

**• p23, L508. Reference the Green Deal.**

Response: Added website link.

**• p23, L513. The sentence starting 'While' ends abruptly, making no sense. Also, rephrase 'whole there here developed methodology'.**

Response: Changed to "While the estimated errors of individual years are larger than those variations, most error components will stay consistent between the years.", changed to "The here developed..."

**• p23, L518. 'tooling' should be 'tools'.**

Response: Adjusted to tool.

**• p23, L52. What is 'link website, mode fields'?**

Response: Removed

**• Fig A1. Be explicit: NOx emissions.**

Response: Added "NOx"

**• p28, Fig A4. Again (as with Fig.5), explain what positive values mean.**

Response: Added "The red values indicate a higher value for the satellite derived emissions compared to the inventory emissions."

**• p30, L562. Mangled NO 2.**

Response: Fixed, also checked adjusted other NO 2 references.

---

## Author Comment (AC2)

We would like to thank reviewer 2 for his/her comments and suggestions.

**Response Reviewer 2**

1. **Satellite images only contain information of NO2, but NOx from the emission inventories includes NO and NO2. In emission inventory, the ratio of NOx to NO2 is different from the ratio in ambient concentration. The conversion of NO to NO2 changes the ratio. The study uses a factor of 1.32 (based on ambient concentrations) for all sources definitely leading to uncertainties.**

   Response:
   The reviewer is correct in stating that the 1.32 factor shows some variability typically depending on the atmospheric concentrations of NO2, NO and Ozone. The choice for 1.32 was based on the value used by Beirle et al., 2019 who in turn based is on the ratio given by Seinfeld and Pandis (2006) over regions under polluted conditions around noontime. Depending on the season and latitude the ratio can shift significantly. More recent studies give values ranging between 1.22 (Riyadh, Beirle et al., 2021, based on modelling) and 1.54 (South Africa, Lange et al., 2022). Furthermore, a study by Griffin et al. (2021) reported NOX:NO2 ratios based on aircraft measurements and model simulations near a biomass burning source, and concluded on a ratio between 1.3-1.5 near the source. To test the representativity for Germany as a whole, we used a simulation with the regional transport model LOTOS-EUROS over 2019 to calculate the NOx:NO2 regions throughout the year for the hours around the TROPOMI overpass (LOTOS-EUROS version 2.2, Manders et al., 2017, more details on simulation on request). The simulated yearly mean averaged values of NOx:NO2 range between 1.3 for northern regions further away from major emissions, and about 1.5 on top of major industrial sources such as power plants and around the more elevated regions. The standard deviation of the daily values (at around the TROPOMI overpass) were also calculated with typical values around 0.1-0.15 and the largest values (<0.3) calculated around the major emission regions (e.g. Powerplants, Ruhr industrial area, Hamburg). The mean values over Germany for 2019 are 1.39 with a standard deviation of 0.16. Both the more recent Beirle et al., (2021) value of 1.41 (for Germany) and our earlier choice of 1.32+-0.26 are within agreement with our simulated results. As variations can be expected from year to year we stick with the earlier value of 1.32 and add the standard deviation of 0.26 to our uncertainty estimate.

[Figure]

Fig. R2.1 Yearly Mean and StDev of NOx:NO2 ratios for 2019.

**Changed lines 285-288** to: "TROPOMI is only capable in observing NO\textsubscript{2}, therefore an additional correction is needed to account for the NO mass. The NOx to NO2 concentration ratio depends on the local chemistry with values commonly falling within the 1.2-1.5 range for polluted regions (Beirle et al., 2011, Beirle et al., 2019, Beirle et al., 2021, Lange et al., 2022}. In this study we apply the 1.32+-0.26 factor as used by Beirle et al., 2019 and include the standard deviation of 0.26 further into the uncertainty budget account for the variations."

Lange et al., 2022 (Lange, K., Richter, A., and Burrows, J. P.: Variability of nitrogen oxide emission fluxes and lifetimes estimated from Sentinel-5P TROPOMI observations, Atmos. Chem. Phys., 22, 2745–2767, https://doi.org/10.5194/acp-22-2745-2022, 2022.)

Beirle, S., Borger, C., Dörner, S., Eskes, H., Kumar, V., de Laat, A., and Wagner, T.: Catalog of $NO_x$ emissions from point sources as derived from the divergence of the $NO_2$ flux for TROPOMI, Earth Syst. Sci. Data, 13, 2995–3012, https://doi.org/10.5194/essd-13-2995-2021, 2021

Manders, A. M. M., Builtjes, P. J. H., Curier, L., Denier van der Gon, H. A. C., Hendriks, C., Jonkers, S., Kranenburg, R., Kuenen, J. J. P., Segers, A. J., Timmermans, R. M. A., Visschedijk, A. J. H., Wichink Kruit, R. J., van Pul, W. A. J., Sauter, F. J., van der Swaluw, E., Swart, D. P. J., Douros, J., Eskes, H., van Meijgaard, E., van Ulft, B., van Velthoven, P., Banzhaf, S., Mues, A. C., Stern, R., Fu, G., Lu, S., Heemink, A., van Velzen, N., and Schaap, M.: Curriculum vitae of the LOTOS–EUROS (v2.0) chemistry transport model, Geosci. Model Dev., 10, 4145–4173, https://doi.org/10.5194/gmd-10-4145-2017, 2017.

2. **Photochemical reactions are different among seasons and day-night. The life time of 4 hours for NO2 uniformly seems unreasonable for all days during 2019-2021. Radiation could be a good indicator for the lifetime.**

Response:

As the reviewer points out the lifetime of NO2 varies throughout the year. The effective lifetime of NOx depends on both the chemical decay rate and loss to surfaces (dry deposition). Of these two the chemical decay is the dominant factor. While radiation can be a good indicator, the lifetime is typically estimated via the availability of OH and production thereof (typically including radiation). Several studies have explored this route before and either estimate the availability of OH by some basic assumptions on production, or by using modelled OH fields (with the drawback of a potential bias within the simulated concentrations). Either route is possible and estimates for the effective lifetimes end up around 2-5 hours for spring and summertime values (Lorente et al., 2019; Valin et al., 2013). Outer estimates for winter are 12-24 hours (Shah et al., 2020).

Several studies report on effective lifetimes derived from fits to observed plumes from cities and large industrial areas. These values typically give a range between 2-5 hours (Goldberg et al., 2021, Fioletov et al., 2022, Beirle et al., 2011,Lange et al., 2022) with a recent study by Fioletov et al. (2022) giving a value of 3.3 hours representative for larger emissions within the US and Canada (2018-2022). Furthermore, Fioletov et al. (2022) also notes that while lifetime has a large impact on the emission estimates, relative changes do not have a major impact when comparing individual years to one another. They point out that 1h deviations from the 3.3 hour mean only changed the emission estimates between years by about 1%.

Besides estimates of lifetime based on observations, we can also look at simulated lifetimes within chemistry transport models. While our LOTOS-EUROS chemistry model has no option to directly write out lifetime, we can look at earlier studies that used a tagging approach to label emissions from individual hours. An earlier study by Curier et al., 2014 did just that to study the source sector contribution of emissions from individual hours to the OMI NO2 column at OMI overpass for several industrial regions in Europe. For the region most representative of Germany (Benelux) the study states: "Approximately 50% of the modelled OMI signal results from NOx emissions in the 3 h prior to OMI overpass.". This statement holds for most the source sectors. Assuming a relatively constant source this translates to a lifetime of about 4 hours (at column level, and assuming basic mass balance).

A potential point of concern remains the representativity for the whole year. Most of the estimates are biased towards spring, summer and autumn as there are typically more observations available within these months. To correct for the representativity bias we already include a seasonal variation factor (1.11), but also remain on the high end of the lifetime estimates by choosing a value of 4.0 hours. The standard deviation of +-1 hour ensure that common values within 3-5 hours remain within the uncertainty range.

**Changed lines 258-259:**

**The effective lifetime of NOx depends on both the chemical decay rate and loss to surfaces (dry deposition). Within our domain of interest the chemical decay will be the dominant factor. Earlier studies using the EMG plume functions derived lifetimes between 2-5 hours based on the decay downwind of major sources worldwide \citep{Beirle_2011, deFoy_2015,Goldberg_2021, Lange_2022, Fioletov_2022}. Following those results we assume a mean lifetime of 4 hours +- 1 hour to account for local and seasonal variations.**

Curier, R.L., Kranenburg, R., Segers, A.J.S., Timmermans, R.M.A. and Schaap, M., 2014. Synergistic use of OMI NO2 tropospheric columns and LOTOS–EUROS to evaluate the NOx emission trends across Europe. Remote Sensing of Environment, 149, pp.58-69.

Fioletov, V., McLinden, C. A., Griffin, D., Krotkov, N., Liu, F., and Eskes, H.: Quantifying urban, industrial, and background changes in NO2 during the COVID-19 lockdown period based on TROPOMI satellite observations, Atmos. Chem. Phys., 22, 4201–4236, https://doi.org/10.5194/acp-22-4201-2022, 2022.

Lorente, A., Boersma, K., Eskes, H., Veefkind, J., Van Geffen, J., De Zeeuw, M., Denier Van Der Gon, H., Beirle, S., and Krol, M.: Quantification of nitrogen oxides emissions from build-up of pollution over Paris with TROPOMI, Scientific reports, 9, 1–10, 2019

Shah, V., Jacob, D. J., Li, K., Silvern, R. F., Zhai, S., Liu, M., et al. (2020). Effect of changing NOx lifetime on the seasonality and long-term trends of satellite-observed tropospheric NO2 columns over China. Atmospheric Chemistry and Physics Discussions, 20(3), 1483– 1495. https://doi.org/10.5194/acp-2019-670

Valin, L. C., Russell, A. R., & Cohen, R. C. (2013). Variations of OH radical in an urban plume inferred from NO2 column measurements. Geophysical Research Letters, 40(9), 1856– 1860. https://doi.org/10.1002/grl.50267.

3. **The comparison of inventory and Tropomi in Figure 3 is not clearly, showing the difference is better for readers.**

   Response: The requested difference plots between the inventory and TROPOMI based emissions are already given in 5A-F.

4. **The abstract needs to be modified as the currently version is not clear about the method and the key results.**

   Response: Rewrote and shortened the abstract to,

"$NO_x$ is an important primary air pollutant of major environmental concern which is predominantly produced by anthropogenic combustion activities. $NO_x$ needs to be accounted for in national emission inventories, according to international treaties. Constructing accurate inventories requires substantial time and effort, resulting in reporting delays of one to five years. In addition to this, difficulties can arise from temporal and country specific legislative and protocol differences. To address these issues, satellite-based atmospheric composition measurements offer a unique opportunity for independent and large-scale estimation of emissions in a consistent, transparent, and comprehensible manner. Here we test the multi-source plume method (MSPM) to assess the $NO_x$ emissions over Germany in the Corona period from 2019-2021. For the years where reporting is available, the differences between satellite estimates and inventory totals were within 75-100 kt ($NO_2$) $NO_x$ (<10% of inventory values). The large reduction of $NO_x$ emissions (~15%) related to the COVID-19 lock-downs was observed in both the inventory and satellite derived emissions. The recent projections for the inventory emissions of 2021 pointed to a recovery of the 2021 emissions towards pre-COVID-19 levels. In the satellite derived emissions however, such an increase was not observed. While emissions from the larger power-plants did rebound to pre-COVID-19 levels, other sectors such as road transport did not, likely due to a reduction in the number of heavier transport trucks. This again illustrates the value of having a consistent satellite based methodology for faster emission estimates to guide and check the conventional emission inventory reporting. The method described in this manuscript also meets the demand for independent verification of the official emission inventories, which will enable inventory compilers to detect potentially problematic reporting issues, bolstering transparency and comparability: two key values for emission reporting."

5. **The discussion of uncertainties is qualitative rather than quantitative. A ranking of uncertainties from different assumptions is helpful for assessing the results when this method is used for other cases.**

Response: The discussion on the uncertainties (lines 430-479) has been moved and extended to form its own section (2.2.2, from line 290 onward) which can be read in the updated version of the manuscript. Additionally a table (1) has been added to summarize the individual uncertainties/errors. Some further explanation has been given for the individual error terms, linking back to earlier sections in the manuscript where needed. The discussion section has also been shortened to account for the moved section.

---

## Author Response (AR2)

*We would like to thank reviewer 1 for his/her constructive comments and suggestions.*

*Major comments:*

**1. Although not brought by the referees originally, the title should mention Germany, since this study only deals with that country.**

Response: Added Germany to the title. New title: "Can TROPOMI-NO2 satellite data be used to track the drop and resurgence of NOx emissions within Germany between 2019 – 2021 using the multi-source plume method (MSPM)?"

**2. Concerning the important questions arising from the 4h lifetime, the authors have made a small adjustment in the text to say 2-5 hours, but my impression is that they are trying to avoid a proper discussion. For example, in their reply to Referee #1 they cite evidence of 12-24 hours in winter, and although this estimate was for China I would expect something like that for Germany in the (high-emitting) winter months.**

**The discussion of footprints in the reply to Ref #2 is also useful, and helpful for readers. Readers of the new manuscript will be unaware of the points made in this reply-to-referees, and will still be surprised I think to see the assumption of 4h without much explanation. The authors should expand this issue in the manuscript, or add much of the reply to referees in the SI.**

Response: We added parts on the discussion on lifetime and footprints/plume spread to the manuscripts:

**Lifetime**, Replaced line 254-259 "The effective … for local and seasonal variations" with: "The lifetime of NO\textsubscript{x} depends on both the chemical decay rate and loss to surfaces (dry deposition). Within our domain of interest, the chemical decay will be the dominant factor. Commonly used lifetimes in literature are typically based on either modelled lifetimes or derived lifetimes from (satellite) observed plumes. Modelled lifetimes are commonly estimated via the availability of OH and production thereof (often including radiation) \citep{ Valin2013, lorente2019quantification}. Several studies have explored this route before and either estimate the availability of OH by some basic assumptions on production, or by using modelled OH fields (with the drawback of a potential bias within the simulated concentrations). Either route is possible and estimates for the effective lifetimes end up around 2-5 hours for spring and summertime values \citep{ Valin2013, lorente2019quantification}. Outer estimates for wintertime lifetimes are 12-24 hours \citep{Shah2020}. Alternatively, lifetimes can be derived from tagging emitted molecules and tracking these within the model domain \citep{Curier2014}. The study reported that for a region representative of Germany (Benelux), approximately 50\% of the modelled satellite signal (Ozone Monitoring Instrument, OMI, \citep{levelt2018}) results from NO\textsubscript{x} emissions in the 3 h prior to OMI overpass.". Assuming a relatively constant source this translates to a lifetime of about 4 hours (at column level, and assuming a basic mass balance). Several other studies report on effective lifetimes derived from fits to observed plumes from cities and large industrial areas. Using the EMG plume functions the studies derived lifetimes between 2-5 hours based on the decay downwind of major sources worldwide\citep{Beirle_2011, deFoy_2015,Goldberg_2021, Lange_2022, Fioletov_2022} with a recent study by \citet{Fioletov_2022} giving a value of 3.3 hours representative for larger emissions within the US and Canada (2018-2022).

Following the modelled and observed lifetimes we assume a mean lifetime of 4 hours $\pm$ 1 hour to account for local and seasonal variations. A potential point of concern remains how representative the lifetime is for the whole year. Most of the estimates are biased towards spring,

summer and autumn as there are typically more observations available within these months. To correct for the representativity bias a seasonal variation factor (1.11) will be included (explained in next section), but additionally by choosing a value of 4.0 hours we remain on the high end of the lifetime estimates. The standard deviation of $\pm$1 hour ensure that common values within 3-5 hours remain within the uncertainty range. Furthermore, \citet{Fioletov_2022} also notes that while lifetime has a large impact on the emission estimates, relative changes do not have a major impact when comparing individual years to one another. They point out that 1h deviations from the 3.3 hour mean only changed the emission estimates between years by about 1\%."

**On the footprints / plumespread: We added more of the discussion from the previous review round,** Replaced line 342-345, "The GNFR data is … source and emission methodology" to "The GNFR data is used as a basis and summed and regridded for all the NFR classes, to match the 0.1ºx0.1º grid used in this study. A Gaussian filter (\textit{scipy.ndimage},\cite{scipy_source}) is applied to the data with a sigma of 1 grid cell. The posteriori smoothing is only there to bridge the limitations of the method and instrument. The spatial limit to resolve 2 sources of similar size depends on the effective lifetime, the pixel size and meteorological factors such as typical diffusion etc. Of these the pixel size and lifetime are dominant at the TROPOMI pixel limit (5.5x3.5km\textsuperscript{2}). The pixel size combined with diffusion gives us a typical plume width of around 7 km (e.g. $\sigma$\textsuperscript{2}=$\sigma$\textsuperscript{2}\textsubscript{plume}+$\sigma$\textsuperscript{2}\textsubscript{pixel}+$\sigma$\textsuperscript{2}\textsubscript{source}). This value varies depending on typical size of a source but most sources of NO\textsubscript{2} are limited in size (except for large mines, very large cities etc). Based on \citet{McLinden:2022} a plume-width of 7 km combined with a lifetime of 4 hours gives an effective resolvability limit of 15-20km, which for 0.1ºx0.1º source cells (e.g. $\sim$10x10km\textsuperscript{2}) explains the choice for a sigma of 1 grid-cell. More smoothing can produce better results, but also reduces the observable details."

*3. p6, L167, and Ref #2 comment. The authors reply that "The MAX-DOAS and PANDORA instruments are not completely free of bias themselves but typically have much lower uncertainties than the TROPOMI-NO2 product as stated in Verhoelst et al., 2021.", but again this information is not passed to the readers of this manuscript. This should be explicit here.*

Response: We added "It is important to note that the MAX-DOAS and PANDORA instruments are not completely free of bias themselves, however the ground-based instruments typically have much lower uncertainties than the TROPOMI-NO2 product as stated in Verhoelst et al., 2021." to Line 176.

*4. Tools and code availability?*

*The footnotes on page 3, and the "Code and data availability" section gives web addresses. I thought GMD insisted on zenodo for software?*

Response: *In my opinion this is generally a point for the typesetting/final editing phase. We can upload a version of the code to Zenodo or any other sharepoint if needed/required.*

*Other points:*

*p11, L294: +a few % plus/minus maybe?*

Response: changed to ±.

*p23, L471. The (Kuenen..) ref should be before the comma.*

Response: reference moved to before the comma.

*We would like to thank reviewer 2 for his/her constructive comments and suggestions.*

**The paper presents a comparison of TROPOMI-based NOx emission estimates and emission inventory data for Germany over the years 2019, 2020, and 2021, including the COVID-19 period. It provides a method to evaluate NOx emission inventories and the possibility of more recent emission information, which can be interesting for emission inventory compilers. The paper is of scientific significance and with the given information reproducible. In some parts, there is a lack of logic, too little explanation, and some misleading discussions. After addressing the comments raised below, the paper should be considered for publication.**

**General comments:**

**For some parts, there is a lack of references. See comments in the specific comment section. Sometimes, statements are given without further or too little explanation. For example, regarding the grid size, the assumption of a 7 km plume spread, the rotation method, the COVID-19 effect, and NOx to NO2 ratios. See comments in the specific comment section. There is quite a lot of discussion (Line 374-399) on the visibility of NOx emissions from highways in the satellite-based maps. However, this is not very well visible to the reader from the mentioned Figures (Fig. 4 and especially not in Fig. 6 showing the differences). I would recommend removing or at least weakening the statements and discussions about the NOx emissions and their changes from highways.**

Response: *Thanks for the detailed feedback. We have incorporated these comments in the manuscripts. We think this gives the reader more context behind the statements made and thereby improved scientific soundness. We also added some additional hedging on the statements related to the traffic emissions.*

**Specific comments:**

**Line 9/10: How do you know the reductions are related to the COVID-19 lockdowns, not the results of political emission reduction strategies?**

Response: *These reductions were simultaneous with lock down measures and breaking with trend lines of declining emissions due to policy measures. Generally speaking this trends don't show reduction of ~15%/y but are more in the order of several precent. We can however never discern between Covid induced changes and other causes. Hence the text was adapted to reflect this.*

**Line 116: Missing reference for: "Globally, the lightning NO constitutes about 3% of the total NOx emission budget."**

Response: *Reference added.*

**Line 152/153: Since this is a more general explanation of the different TROPOMI products (NRTI, OFFL, RPRO), the given years (2019-2021 and April 2018-November 2018) are confusing. Since they are also not used/relevant for the study, I suggest to remove them. Also, delete the word "reprocessed" in combination with the offline mode in line 152.**

Response: *It is true that another data product was used (PAL). This was however added to stress there are multiple versions of TROPOMI NO2 and this topic of continuous improvement. This is now framed differently and more clearly. Still we think it is useful to mention all options here.*

*Line 153: At this point, there is no explanation given why a reprocessed version is needed. Please add something like: "Over time, several modifications in the retrieval lead to processor updates and new product versions. Reprocessed data sets based on the latest offline version are provided at a more irregular interval."*

Response: *Thanks for this suggestion. It has been incorporated (with slight adaptations) in the manuscript.*

*Line 156-159: Relevant for this study in this detail? Explanation would have been helpful in line 153 (see comment above). Maybe move some parts of it or delete it.*

Response: *We think it is good to keep this overview. The reader should be aware which datasets are used in order to guarantee reproducibility. Knowing there are many data products and version is crucial in that regard.*

*Line 173: Reference missing for "The negative bias can be explained by the low spatial resolution of the a-priori profiles as well as the treatment of clouds and aerosols in the retrieval." Is this shown in van Geffen et al. (2022), otherwise, add Lange et al. (2023, https://doi.org/10.5194/amt-16-1357-2023).*

Response: *Lange et al was added here.*

*Line 173: Sentence difficult to understand. What is meant by set? Where does this recommendation of 0.3 come from? I am only familiar with the cloud radiance fraction filter of 0.5, which is also the filter when the qa_value of 0.75 is applied.*

Response: *The sentence was rephrased. The requirements came from the ATBD (algorithm theoretical basis document) for this data product this is now more clear. The 0.3 cloud radiance fraction filter requirement comes from earlier assimilation applications.*

*Line 177: How do you know that it is COVID-19 impact, not meteorology or reducing strategy (see, for example, Goldberg et al. (2020, https://doi.org/10.1029/2020GL089269)) especially since it is not coming back in 2021? See also comment regarding line 9/10.*

Response: *See response to the comment regarding line 9/10. Here also the phrasing in the manuscript was adapted.*

*Line 193: I think the word "tooling" is not always used correctly; maybe better to say "Emission estimation tool". See also line 196.*

Response: *Thanks for this suggestion. The change is incorporated in the manuscript*

*Line 195: The term "space-emissions tool" is misleading. Better would be a "satellite-based emissions tool".*

Response: *That is more accurately put. We adapted the text accordingly.*

*Line 195-200: Repetition this was already discussed in the introduction. Since it is not very relevant for the paper, it should be deleted in the introduction.*

Response: We think this is provides valuable context for the developments presented in the manuscript and hence we decided to keep it. It is however true that this is somewhat repetitive. Hence we shortened the text and removed the repetition.

*Line 206: Confusing: With "here" you mean the tool and not the paper? So, the tool uses the mass-balance technique, but the paper is focused on the plume based fitting method.*

Response: *Yes, this is now clarified further in the manuscript.*

*Figure 2 legend: "Space-emissions tool" see above, Replace "A" with "Panel A".*

Response: *Changed accordingly*

*Line 222 and 223: The term "total column density" is confusing since it is also the term for the column from the surface to the top of the atmosphere, but I think this is not meant here.*

Response: *That is meant here. Or more precisely tropospheric column density. Text was changed accordingly.*

*Line 235: "Using this method, observations are rotated around a single point, the emission source, so that each is positioned in a similar upwind-downwind frame" How do you know your emission sources, what about area sources like highways?*

Response: This is a good question and it should be seen as a potential source. We don't know the sources yet and that is why a grid of potential emission locations is introduced . The strength of the method is that its independent of the initial emission locations. This is now clarified in the text.

Highway sources can be seen as a continuous line of point sources instead of an area source, which aggregates down to 0.1x0.1 degree steps. A similar thing can be said for area sources. See the discussion on resolvability as to why the limitation of a 0.1x0.1 degree grid is valid. Alternatively one could describe highways as a more detailed set of single source points with a single enhancement multiplier for segments of the road to increase resolvability. This was however not the goal of the study.

*Line 251/252: Change "by dividing the emission enhancement ai by the decay-rate $\lambda$;" to "by dividing the emission enhancement ai by tau".*

Response: Agreed and incorporated. Thanks for noticing this error.

*Line 253: What are the reasons for a grid with a resolution of 0.1 x 0.1 degrees? How does the resolution influence the results? Please provide a short statement.*

Response: *The resolution is chosen as a compromise between computational burden, level of detail required, limitations of the instrument (resolvability) and conditioning of the linear system describing the source receptor relations. This is added to the text.*

*Line 255: "Within our domain of interest, the chemical decay will be the dominant factor." How do you know?*

Response: We have a lot of experience in modelling NOx with chemical transport models in which case you can determine the strength of the loss terms (wet- and dry-deposition, chemical sinks etc) because you can track budgets for the various processes. Chemical processes dominate the loss of NOx within this domain.

*Line 261: 7 km of plume spread seems small to me, isn't the plume spreading over a larger area quite quickly due to diffusion? It would be good to add references to other studies using similar values.*

Response: Other studies by Griffin et al., and Fioletov use 7 and 8km. In both cases the authors derived the plume width from fits to single sources. The plume width itself has a fairly limited result on the resulting emissions. These references are added.

**Line 265/266: Why do you need a correction factor for seasonal variability when you can get seasonal information from satellite observations?**

Response: Seasonal was removed. This is indeed not required. When averaging one needs to take into account that there are more observations in summer than in winter generally.

**Line 268: Please comment on whether assuming a constant lifetime over the day and for different seasons is valid.**

Response: A similar question came from the earlier(other) reviewer(s), see those replies for further explanation. We added to the manuscript the following "commonly used lifetimes in literature are typically based on either modelled lifetimes or derived lifetimes from (satellite) observed plumes. Modelled lifetimes are commonly estimated via the availability of OH and production thereof (often including radiation) \citep{ Valin2013, lorente2019quantification}. Several studies have explored this route before and either estimate the availability of OH by some basic assumptions on production, or by using modelled OH fields (with the drawback of a potential bias within the simulated concentrations). Either route is possible and estimates for the effective lifetimes end up around 2-5 hours for spring and summertime values \citep{ Valin2013, lorente2019quantification}. Outer estimates for wintertime lifetimes are 12-24 hours \citep{Shah2020}. Alternatively, lifetimes can be derived from tagging emitted molecules and tracking these within the model domain \citep{Curier2014}. The study reported that for a region representative of Germany (Benelux), approximately 50\% of the modelled satellite signal (Ozone Monitoring Instrument, OMI, \citep{levelt2018}) results from NO\textsubscript{x} emissions in the 3 h prior to OMI overpass.". Assuming a relatively constant source this translates to a lifetime of about 4 hours (at column level, and assuming a basic mass balance). Several other studies report on effective lifetimes derived from fits to observed plumes from cities and large industrial areas. Using the EMG plume functions the studies derived lifetimes between 2-5 hours based on the decay downwind of major sources worldwide\citep{Beirle_2011, deFoy_2015,Goldberg_2021, Lange_2022, Fioletov_2022} with a recent study by \citet{Fioletov_2022} giving a value of 3.3 hours representative for larger emissions within the US and Canada (2018-2022)." on the discussion of lifetime was added. We think this point is now covered amply.

**Line 271: What is the temporal resolution of these profiles? Are there individual profiles for the different sources available, if yes, how large is the variability in the correction factor? Is it valid to use an averaged profile over all sources?**

Response: Hourly by default and yes there are different time profiles for different sources. The variability differs per source. Traffic for example has stronger fluctuations and certain industries are simply flat. Using an averaged profile is always an assumption that potentially introduces a misrepresentation at specific locations.

**Line 286: The NOx to NO2 ratio is influenced by seasonality and time of the day. At least add something like: The NOx to NO2 concentration ratio depends on the local chemistry, influenced by ozone concentration, photolysis frequency of NO2 (solar zenith angle), and the rate constant of NO+O3 reaction (temperature).**

Response: Added.

*Line 287: How is the value of +/- 0.26 determined, is it really the standard deviation, how do you know the distribution of the NOx to NO2 ratios?*

Response: This can be estimated using CTM output or measurements. Both of these approaches come with uncertainties. Generally values on the standard deviation of ~20% come out of these studies. This value was also used in Beirles 2021 paper.

*Line 300: Please provide a reference, especially for the large part regarding the missing variations in the stratospheric NO2 concentration; the first part was mentioned before.*

Response: Reference is added.

*Line 325: You only discuss the wind speed, what is about wind direction, which is essential for the rotation?*

Response: Yes. +-1 ms-1 in eastward and northward directions, its implicitly included. The text is now reformulated to stress this more.

*Line 374: I only see elevated NOx emissions from the Rhine-Ruhr region to Hannover but not between Hannover and Magdeburg and not towards Berlin. At least not in the satellite-based maps. There are also several large power plants close to highway A2 near the Ruhr area, which is visible in your Fig. 6, so I probably don't see the NOx emissions from the highway but from the power plants located nearby. I would recommend removing this part about the highways.*

Response: We have to disagree with this statement. It might require a keen eye but we are confident the road network (an particularly the busies highways) are visible in the emission estimate. We do appreciate that one needs to have a thorough look and hence some hedging of the statements concerning road networks was added.

*Line 394: You mention a noticeable drop in emissions around highways. Where I think this is not visible. There is probably a reduced signal in the region of the A2 highway close to the Ruhr area, but this isn't easy to disentangle since there are several power plants along the highway in this part. See comment regarding line 374. I would recommend removing the part about the highways here.*

Response: See the response to the previous comment. We agree that at the location of the A2 highway we see a clear drop but also we are confident the signals from the A1 and A3 are visible. Actually plotting the road network on the emission estimates might show this more clearly to the reviewer:

[Figure]

*Line 396: You mention a rise in emissions from 2020 to 2021, most noticeable in large urban areas. I think this is very hard to see...maybe Berlin shows higher emissions in 2021 compared to 2020, but I don't see any clear rise in signal. Please provide examples, including values for the rising emissions.*

Response: We added a refence to figure 6 in the text here that shows this more clearly.

*Line 398/399: The Hannover-Rhine-Ruhr part is still visible in 2020, but especially the part with large emitters (see Fig. 6) along the highway. All this highway discussion is on very small differences and not really visible in the provided maps.*

Response: We agree that the plots are small as presented in the text. However they have a high resolution and remain of decent quality when one zooms in to be able to appreciate these changes that are indeed quite nuanced. It is however required to do this because at first glance differences are not always apparent.

*Line 415: "The Weisweiler power-plant reduces into 2019 while reducing further into 2021." I cannot follow this statement; do you mean 2020 instead of 2019?*

Response: Yes! Thanks for pointing this out.

*Figure 8 legend: "the red error bars show the uncertainty in the satellite derived" The -50 to +30% given in Table 1? But the error bars look similar in + and - directions? This national inventory shows higher emissions in 2021 than in 2020, especially from road transport. In line 407, you stated an "upsurge in the usage of coal-fired power stations for power generation in 2021 compared to the COVID-19 year 2020". I think this is not visible here, do you have an explanation?*

Response: You do see a slightly higher Public power contribution in 2021 compared to 2020. Maybe upsurge is a bit of an exaggeration. We have changed the text accordingly.

*Line 435: What is meant by power generation, public power, or industry? Probably public power, but also for public power, your Fig 9. upper left panel shows an increase in 2021 (0.925) compared to 2020 (0.85), this is not almost back to 2019, it is in the middle between the two years.*

Response: Correct. The is now altered in the text

*Line 438/439: "While Road transport emissions were expected to show a recovery this is not matched by patterns in the satellite derived emissions." Your Fig. 9 shows that road transport emissions decreased from 2019 to 2020, but satellite data show no change from 2020 to 2021. However, this reduction from 2019 to 2020 is not visible from Fig. 6, as discussed earlier. How can you explain the apparent difference between the discussed maps and Fig. 9?*

Response: This might have to do with the other sources present in the regions that are labelled as road transport dominated (above 50% of the total emissions from road transport). This means that up to maximally half of the emission can originate from another sector that can challenge drawing conclusion. By the way, in figure 6 we do see a reduction in emissions from road traffic as light blue patches co-located with the road network.

*Line 440/441: Not so easy to say for shipping emissions: The inventory shows a recovery. Why the difference here? However, industry emissions have continued their downward trend with no sign of recovery but are not mentioned here. In general, it is difficult to speak of a trend when it is only two years.*

Response: That is a valid point. We rephrased this. Concerning the recovery in the emission inventory we know how challenging it is to accurately incorporate shipping emission into an emission inventory. How the post covid recovery is taking into account requires detailed knowledge from the compilers of these emission inventories hence we cannot answer this question confidently.

*Figure 9 legend: "A clear downward trend is visible for most sectors." Not really true, maybe for industry, and in general, it is difficult to speak of a trend when it is only two years. I would suggest deleting the sentence.*

Response: Agreed. Text has been adjusted.

*Line 471: This is not clear to me: emissions 2018, based on the in 2020 reported emissions*

Response: For emissions there are always two date to take into account. One the reporting year and one is the year of the emissions. So this means 2018 emission as reported in 2020. The text has been adjusted.

*Line 552/553: How will this work with the worse resolution of OMI compared to TROPOMI?*

Response: Since we are aggregating to 0.1x0.1 degree (~6x11km) the OMI resolution (13 km × 24 km) will start hampering the level of detail available in the map. It might be more optimal to switch to a 0.2x0.2 grid for OMI as the limited footprint will probably not provide enough information to prevent overfitting within a 0.1x0.1 degree grid. This is now also mentioned in the text.

*Technical corrections:*

*Line 30: Change NO2 emissions to NOx emissions*

Response: changed to NOx

*Line 39/40: Two times "for example", please rephrase the sentence*

Response: Change "For example, emissions from road transport for example depend on several factors such as:" to "For example, emissions from road transport depend on several factors such as"

**Line 45/46: Change "For example air quality applications such as forecasts" to "For example, air quality forecasts"**

Response: changed as requested.

**Line 55: x of NOx is not subscript**

Response: Changed to subscript.

**Line 55/56: Change "Timely verification of the inventories could potentially more rapidly identify such discrepancies." to "Timely verification of the inventories could potentially identify such discrepancies more rapidly."**

Response: Changed as requested.

**Line 58/59: Change to "Furthermore, due to increased instrument sensitivity and spatial and temporal resolution, these satellite-based measurements ..."**

Response: Changed as requested.

**Line 88: Change "space-borne measurement" to "space-borne emission estimate"**

Response: Changed as requested.

**Line 89/90: Change "In this study we focus on the plume based fitting method." to "In this study, we focus on the results of the plume-based fitting method."**

Response: Changed as requested.

**Line 106: Change "Agricultural Soils" to "agricultural soils"**

Response: Changed as requested.

**Line 107: delete "inventory", it is doubled with the following "reported emission data sets" and "inventories" in line 108**

Response: Changed as requested.

**Line 117/118: Why twice reference to EEA (2019)? The first one is enough.**

Response: Changed as requested.

**Line 125: Change "as kt (NOx)" to "as kt NOx"**

Response: Changed as requested.

**Line 128: Sentence not complete.**

**..., mostly based on the anthropogenic emissions Yienger and Levy II (1995) reported, ...**

**or**

**..., mostly based on the anthropogenic emissions (Yienger and Levy II, 1995), ...**

Response: changed to "*mostly based on the anthropogenic emissions Yienger and Levy (1995) reported*"

***Line 152: Space missing between offline and (OFFL)***

Response: Changed as requested.

***Line 153: Delete "Finally"***

Response: Changed as requested.

***Line 183: Chance section from "Meteorology" to "Wind data"***

Response: Changed as requested. Line 185 also adjusted "the required meteorology" to 'the required wind data"

***Equations 4 and 6: Functions (exp, erfc) should not be written in italics***

Response: Changed as requested.

***Line 261: Change "equation" to "Equation"***

Response: Changed as requested.

***Line 307: Change "Divergence" to "divergence"***

Response: Changed as requested.

***Line 320: Change "seems to be" to "is"***

Response: Changed as requested.

***Line 358: Change "2d" to "2D" as in line 356.***

Response: Changed as requested.

***Line 369: Repetition of "emissions", remove "emissions" after inventory.***

Response: Changed as requested.

***Line 371: Change "or" to "respectively"***

Response: Changed as requested.

***Line 382: Add: …such as power plants, visible in the top row without the Gaussian filter.***

Response: Changed as requested.

***Line 382: "with a large spatial footprints": remove "a"***

Response: Changed as requested.

***Line 383: Change "satellite instrument" to "TROPOMI's spatial resolution"***

Response: Changed as requested.

***Figure 6 legend: Change "2020-2019 and 2021-2019" to "***2020 and 2019, respectively 2021 and 2019***"***

Response: Changed as requested.

***You write Power-Plants and powerplants in the legend of Fig. 6. It's power plants. Please check throughout your text. Sometimes you have power-plant or powerplant.***

Response: Changed as requested.

*Figure 7 legend: Change "2020-2019 and 2021-2019" to "2020 and 2019, respectively 2021 and 2019" There are only 2 panels, change 3 to 2.*

Response: Changed as requested.

*Line 423: Change "2019-2020" to "2019 to 2020"*

Response: Changed as requested.

*Line 429: Somewhere, name the 5 sectors, maybe just in brackets: Only 5 sectors (public power, industry, ...) have locations ...*

Response: added as requested,  "Only 5 sectors (public power, industry, road transport, shipping and agriculture other than livestock)…"

*Line 430, 431, 432, 433: Change PublicPower, Shipping, … to public power and shipping*

Response: changed as requested

*Line 438: Change "Road" to "road"*

Response: changed as requested

*Line 440: 3500 km instead of kg?*

Response: no correct as stated, it means trucks above 3500kg. Relatively standardized weight above which a different driver license is needed throughout most of the EU. Added "(vehicles with a weight above >3500kg)" for clarification.

*Line 444: Change "in inventory emissions" to "the analyzed inventory emissions"*

Response: changed as requested.

*Line 445: Unit missing 75-100 kt NOx*

Response: changed to "kt NOx"

*Line 447: What is meant by series and sets? Maybe better both emission estimates*

Response: changed as requested.

*Line 447: Move reference to Fig. 8 forward by one sentence*

Response: shifted forward by one sentence

*Line 454: Add: Additionally, this approach…*

Response: changed as requested.

*Figure 9 legend: Change other line to solid line*

Response: changed as requested

*Line 465: What is meant by inversions here? I think this is not the right word.*

Response: changed "inversions " to " satellite derived emissions"

*Line 466: Change "The stronger the source is, the better is the resolvability." to "The stronger the source, the better the resolvability"*

Response: changed as requested

*Line 552: Change "In future" to "In the future" and delete "also"*

Response: changed as requested